# CHD3 helicase domain mutations cause a neurodevelopmental syndrome with macrocephaly and impaired speech and language

Lot Snijders Blok, Justine Rousseau, Joanna Twist et al.[#]

Chromatin remodeling is of crucial importance during brain development. Pathogenic alterations of several chromatin remodeling ATPases have been implicated in neurodevelopmental disorders. We describe an index case with a de novo missense mutation in *CHD3*, identified during whole genome sequencing of a cohort of children with rare speech disorders. To gain a comprehensive view of features associated with disruption of this gene, we use a genotype-driven approach, collecting and characterizing 35 individuals with de novo *CHD3* mutations and overlapping phenotypes. Most mutations cluster within the ATPase/helicase domain of the encoded protein. Modeling their impact on the three-dimensional structure demonstrates disturbance of critical binding and interaction motifs. Experimental assays with six of the identified mutations show that a subset directly affects ATPase activity, and all but one yield alterations in chromatin remodeling. We implicate de novo *CHD3* mutations in a syndrome characterized by intellectual disability, macrocephaly, and impaired speech and language.

The Chromodomain Helicase DNA-binding (CHD) protein family is a key class of ATP-dependent chromatin remodeling proteins, which utilize energy derived from ATP hydrolysis to regulate chromatin structure, thereby modulating gene expression[1,2]. CHD proteins are crucial for developmental processes[1,3], with various members implicated in major neurodevelopmental disorders including CHD2 in epileptic encephalopathy[4], CHD7 in CHARGE syndrome[5], CHD8 in autism[6,7], and more recently CHD4 and CHD1 in neurodevelopmental syndromes[8,9]. Three CHD proteins (CHD3, CHD4, and CHD5) can exert their chromatin remodeling activity by forming the core ATPase subunit of the NuRD complex[1,10–12]. The NuRD complex is associated with various fundamental cellular mechanisms, including genomic integrity and cell cycle progression[13], and plays important roles in embryonic stem cell differentiation[14]. A recent study reports that the different CHD factors within the NuRD complex (CHD3, CHD4, and CHD5) are developmentally regulated in the mouse brain, each having distinct and mostly non-redundant functions during cortical development[15]. In particular, the CHD3 protein has been implicated in late neural radial migration and cortical layer specification.

In contrast to most other members of the CHD protein family, a specific syndrome associated with mutations in *CHD3* (MIM 602120) has not yet been characterized. In this study, based on an index case from whole genome sequencing of children with rare speech disorders, we assemble a set of 35 probands carrying de novo mutations that disrupt *CHD3*. We characterize the overlapping phenotypic features of probands with *CHD3* mutations, including intellectual disability (with a wide range of severity), developmental delays, macrocephaly, impaired speech and language skills, and characteristic facial features. We identify mainly missense mutations that cluster in and around the ATPase/helicase domain of the CHD3 protein, and are predicted to disturb function, based on three-dimensional modeling. We use functional assays to describe the effects of multiple different CHD3 mutations on ATPase activity and chromatin remodeling capacities. Taken together, our data demonstrate that de novo missense mutations in *CHD3* disturb chromatin remodeling activities of the encoded protein, thereby causing a neurodevelopmental disorder.

## Results

**De novo CHD3 mutations cause a neurodevelopmental phenotype.** During whole genome sequencing of a cohort of 19 unrelated children with a primary diagnosis of Childhood Apraxia of Speech (CAS)[16], we discovered a de novo missense mutation in *CHD3*, predicted to disrupt the helicase domain of the encoded protein. CAS is a rare neurodevelopmental disorder characterized by impairments in learning to produce the coordinated sequences of mouth and face movements underlying fluent speech. Remarkably, the CHD3 protein is one of the few documented interaction partners of FOXP2 (see Supplementary Table S1 in ref. [17]), a transcription factor that has been implicated in monogenic forms of CAS, accompanied by wide-ranging language problems, in multiple families and unrelated cases[18–20].

Discovery of the *CHD3* mutation (NM_001005273.2, p. Arg1169Trp) in our index case motivated a search for other de novo mutations in this gene. Studies of large numbers of simplex families with an autistic proband have documented just two single non-synonymous de novo variants in *CHD3* in probands[21,22], while eight additional non-synonymous variants were recently recorded in a study of thousands of children with unexplained developmental disorders from the UK[23], with limited information on phenotypic profiles of carriers of *CHD3* variants. Via GeneMatcher[24] we assemble a cohort of 35 independently

diagnosed probands with de novo mutations disrupting *CHD3*, to systematically assess the phenotypic consequences of damage to this gene.

The 35 probands with de novo mutations in *CHD3* show overlapping phenotypes, summarized in Table 1 and in more detail in Supplementary Data 1. All individuals have global developmental delays and/or intellectual disability, with a total IQ varying from 70–85 (borderline intellectual functioning) to below 35 (severe intellectual disability). Nine individuals (29%) show autism or autism-like features, including stereotypic and hand-flapping behavior. Interestingly, the majority of individuals (19 individuals; 58%) have macrocephaly, and in cases where neuroimaging has been performed, widening of cerebrospinal fluid spaces is noted in 10 out of 30 MRI reports (33%). One individual (individual 5) has microcephaly. Hypotonia is reported in 21 individuals (75%). The facial phenotype consists of widely spaced eyes, a broad and bossing forehead, periorbital fullness and narrow palpebral fissures, laterally sparse eyebrows, low-set and often simple ears with thick helices, and a pointed chin (Fig. 1). Joint dislocations and/or hyperlaxity are reported in 12 cases, and five individuals have inguinal or umbilical hernias. Five of the 21 male individuals have undescended testes. Vision problems are quite common and include hypermetropia (11 individuals), strabismus (10 individuals), and cerebral visual impairment (three individuals). One individual (individual 34) developed epilepsy, two additional individuals had neonatal convulsions. In many individuals an abnormal and often

**Table 1 Summary of phenotypes found in this cohort of probands with *CHD3* mutations**

|  | Amount | Percentage |
|---|---|---|
| **Development** | | |
| ID/DD | 35/35 | 100% |
| *Degree of ID/DD* | | |
| Borderline ID | 3/35 | 9% |
| Mild or mild–moderate ID | 9/35 | 26% |
| Moderate or moderate–severe ID | 8/35 | 23% |
| Severe ID | 7/35 | 20% |
| DD/level unknown | 8/35 | 23% |
| Speech delay/disorder | 33/33 | 100% |
| Autism or autism-like features | 9/31 | 29% |
| **Neurology** | | |
| Hypotonia | 21/28 | 75% |
| Macrocephaly | 19/33 | 58% |
| Widened CSF spaces (MRI) | 10/30 | 33% |
| Neonatal feeding problems | 10/32 | 31% |
| **Dysmorphisms** | | |
| High, broad, and/or prominent forehead | 28/33 | 85% |
| Widely spaced eyes | 24/31 | 77% |
| **Other** | | |
| Joint laxity (generalized and/or local) | 12/30 | 40% |
| *Vision problems* | | |
| Hypermetropia | 11/29 | 38% |
| Strabism | 10/33 | 30% |
| Cerebral visual impairment | 3/33 | 9% |
| Genital abnormalities in males | 6/17 | 35% |
| Hernia (inguinal, umbilical, hiatal) | 5/28 | 18% |

More extensive clinical information per individual is provided in Supplementary Data 1. As information on the different features was not always applicable or known for each patient, the denominator in the "Amount" column is different for different clinical characteristics

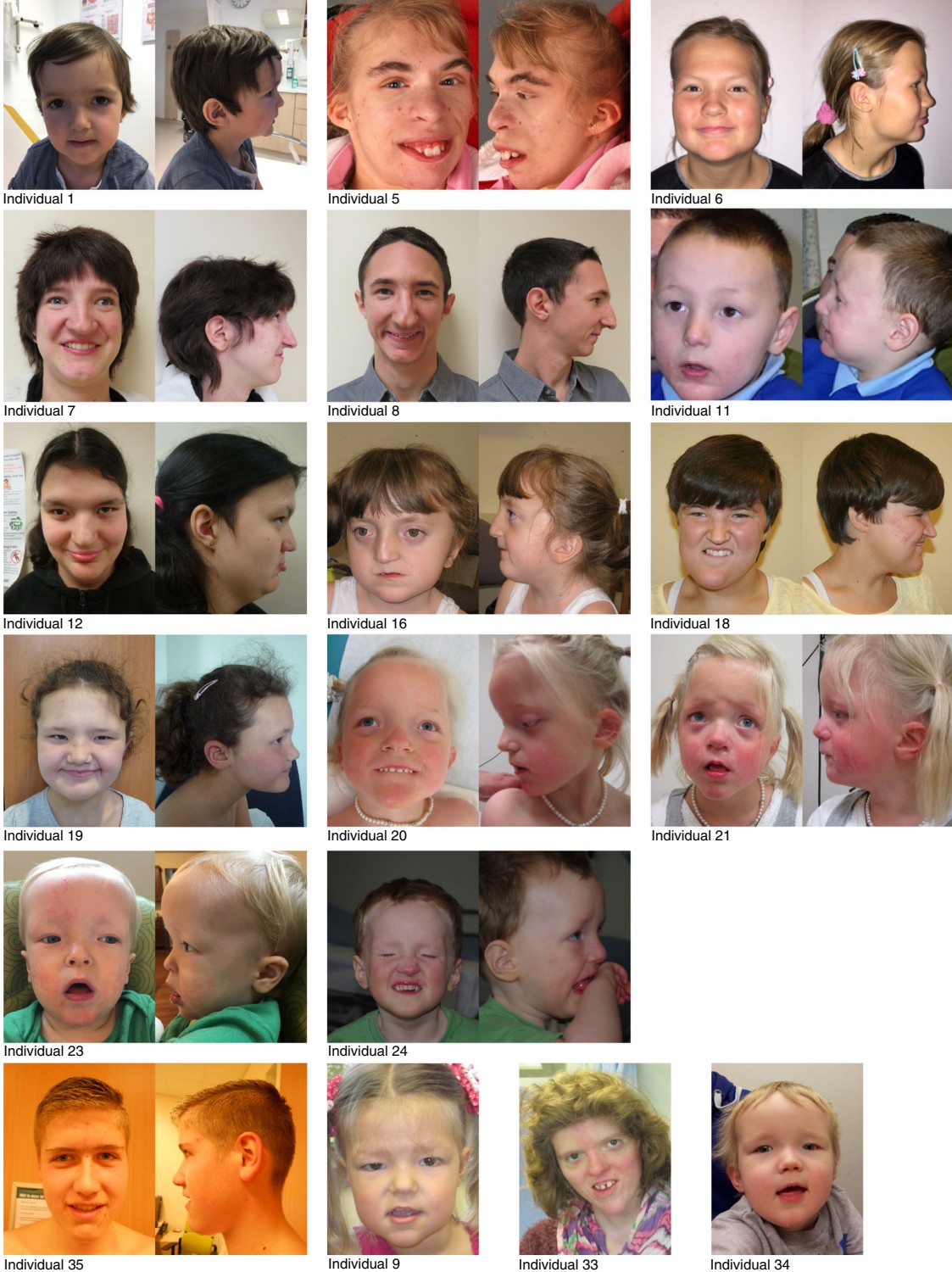

**Fig. 1** Photographs of affected individuals. Facial photographs showing dysmorphisms in 18 individuals with de novo *CHD3* mutations. The majority of individuals have macrocephaly with a prominent or bossing forehead, individual 5 has microcephaly. Hypertelorism or telecanthus is common, often accompanied by narrow palpebral fissures, deep-set eyes, peri-orbital fullness, and/or epicanthal folds. The combination of macrocephaly and deep-set eyes leads to a more prominent supra-orbital ridge. Some individuals show midface hypoplasia. Many individuals have low-set ears that can be posteriorly rotated, and sometimes simple with thick helices. A broad nasal base, prominent nose, a bifid nasal tip, and characteristic pointy chin is also frequently seen, as well as laterally sparse eyebrows

unsteady gait is reported, and one individual (individual 13) developed symptoms of Parkinsonism at a later age.

Given that our index case was ascertained on the basis of a formal diagnosis of CAS, we pay special attention to the association of *CHD3* mutations with speech and language deficits. The index case was diagnosed with severe speech apraxia at the age of 3 years, and then used sign language to communicate effectively. He has severe problems with expressive speech, against relatively normal scores on language comprehension tests and a composite IQ (KBIT) of 72. In all 33 subjects that were at least 2 years old at the last evaluation, *CHD3* disruptions are associated with delayed milestones in the speech and language domain. The average age for first spoken words in this cohort is 2 years and 10 months (range: 1.5–5.5 years, after excluding six individuals that were non-verbal at the last evaluation). Our data suggest that expressive language is more affected than receptive language, and intelligibility is often impaired. Speech-related problems identified in our cohort include dysarthria, speech apraxia, oromotor problems, and stuttering.

**De novo CHD3 mutations cluster in the helicase domain**. The 35 unrelated probands have 23 different de novo mutations in *CHD3* (Fig. 2a, b). None of these mutations are present in the GnomAD database (http://gnomad.broadinstitute.com). Except for four individuals, all individuals have missense mutations. Interestingly, within our cohort there are multiple cases of recurrent identical de novo mutations, revealing mutational hotspots. The most striking is p.Arg985Trp, found in six children from five different families, while two additional individuals have a different substitution affecting the same residue (p.Arg985Gln).

The *CHD3* protein is characterized by a SNF2-like ATPase/helicase domain, together with two plant homeodomain (PHD) fingers and two chromodomains (Fig. 2b, c)[1,11], which mediate chromatin interactions and nucleosome remodeling[1]. The overwhelming majority of missense mutations (17/19) cluster within and around the ATPase/helicase motif, a functional domain that consists of two subdomains: a Helicase ATP-binding lobe and a Helicase C-terminal lobe. This domain provides energy for nucleosome remodeling through its ATPase activity. All missense mutations affect amino acids that are highly conserved, both in different species and also in the other CHD proteins that can be part of the NuRD complex (Supplementary Fig. 1), and clearly cluster in and around highly conserved SF2-family helicase motifs (Supplementary Fig. 1). All are predicted to be pathogenic by Polyphen-2 and/or SIFT, and have CADD scores above 24 (Supplementary Data 1).

The identified de novo mutations also include one in-frame deletion of one amino acid (p.Gly1109del) and two truncating mutations (p.Glu457* and p.Phe1935Glufs*108), although the latter causes a frameshift at the very end of the protein, leading to a stop codon after 108 amino acids. RNA sequencing of transcripts with and without cycloheximide showed that this mutation escapes nonsense-mediated decay (Supplementary Fig. 2). Finally, one case has a splice-site mutation (c.4073-2A>G) which is predicted to yield skipping of exon 27, while preserving the reading frame (Fig. 2a). Data from the ExAC database (http://exac.broadinstitute.com) indicate that *CHD3* is extremely intolerant for loss-of-function mutations (loss-of-function intolerance score of 1.0) and highly intolerant for missense mutations (Z-score of +7.15)[25], supporting the pathogenicity of the mutations that we found.

All *CHD3* mutations were determined to be the most likely causal variant contributing to the disorder of the proband. In proband 15 who has a de novo *CHD3* p.Asp1120His mutation, a de novo truncating mutation in *CIC* was also identified (NM_015125.3:c.1444G>T; p.Glu482*). Since truncating mutations in *CIC* were recently suggested as a potential cause of intellectual disability (ID)[26], both mutations might be involved in the phenotype of this proband.

**A subset of CHD3 mutations directly affects ATP hydrolysis**. The striking clustering of almost all missense mutations in the ATPase/helicase domain of the CHD3 protein led us to hypothesize that disturbance of ATPase and/or chromatin remodeling activities of CHD3 could be potential pathogenic mechanisms. Three-dimensional modeling and mutation analysis of all missense mutations, including analysis of the conserved SF2-characteristic helicase motifs, demonstrates clear clustering of mutations and disturbance of important binding and interaction domains (Fig. 2d and Supplementary Note 1). Direct fluorescence imaging of mCherry-tagged CHD3 mutations in cellular models revealed no differences in subcellular localization for the mutated proteins as compared to wild-type CHD3 (Supplementary Fig. 3).

We experimentally assessed ATPase activity of six representative mutations, selected to include one mutation in the Helicase ATP-binding lobe and several mutations in the Helicase C-terminal lobe. FLAG-tagged full-length wild-type CHD3 protein and each of the six mutant proteins were transiently expressed in mammalian HEK293 cells and purified (Supplementary Fig. 4). Radiometric ATPase assays were performed to assess the activity of these mutant proteins relative to wild-type, in the presence of dsDNA (Fig. 3), recombinant nucleosomes (Fig. 3), or in the absence of DNA substrates as a control (Supplementary Fig. 5). ATPase activities of p.Arg1121Pro and p.Arg1172Gln were significantly lower than wild-type for both substrate conditions. These findings are consistent with the modeling data, since p.Arg1121Pro is predicted to disrupt a helix integral to motif V, while p.Arg1172Gln is located in helicase motif VI, and both motifs are known to be critical in ATP hydrolysis. The activity of p.Asn1159Lys was significantly lower only in the presence of dsDNA, although the reason for the different activity depending on the substrate is currently unknown. The protein with the p.Leu915Phe mutation, located in conserved SNF2-motif III, is significantly hyperactive under both conditions. The p.Arg1187-Pro and p.Trp1158Arg mutations do not show statistically significant differences from the wild-type protein in these ATPase assays. According to the three-dimensional structure, the location of p.Arg1187Pro is not close to the ATP-binding or interaction surface. To assess whether mutant protein could impact activity of wild-type enzyme, we mixed wild-type protein with equimolar amounts of several mutant proteins, finding no biochemical evidence in this assay for interference (Supplementary Fig. 6).

**CHD3 mutations disturb chromatin remodeling capacities**. We measured the effects of six mutations on the chromatin remodeling activity of CHD3, by assessing restriction enzyme accessibility to nucleosomal DNA[27]. Consistent with its reduced activity in the ATPase assays, the p.Arg1172Gln mutant was partially, but not fully, active at chromatin remodeling (Fig. 4). p.Arg1121Pro, which showed severely reduced ATPase activity, was highly compromised in the chromatin remodeling assay. Moreover, p.Leu915Phe demonstrated hyperactivity in this assay, mirroring its elevated ATPase activity. Crucially, chromatin remodeling assays can also detect functional defects beyond ATP hydrolysis[27]. Two of the mutant proteins, p.Trp1158Arg and p.Asn1159Lys, exhibited severely compromised ability to remodel chromatin (Fig. 4) against a background of some preserved ATPase activity (c.f. Fig. 3). In sum, with the sole exception of p.Arg1187Pro, all the mutant versions of CHD3 that we tested differ from wild-type protein in their ability to remodel

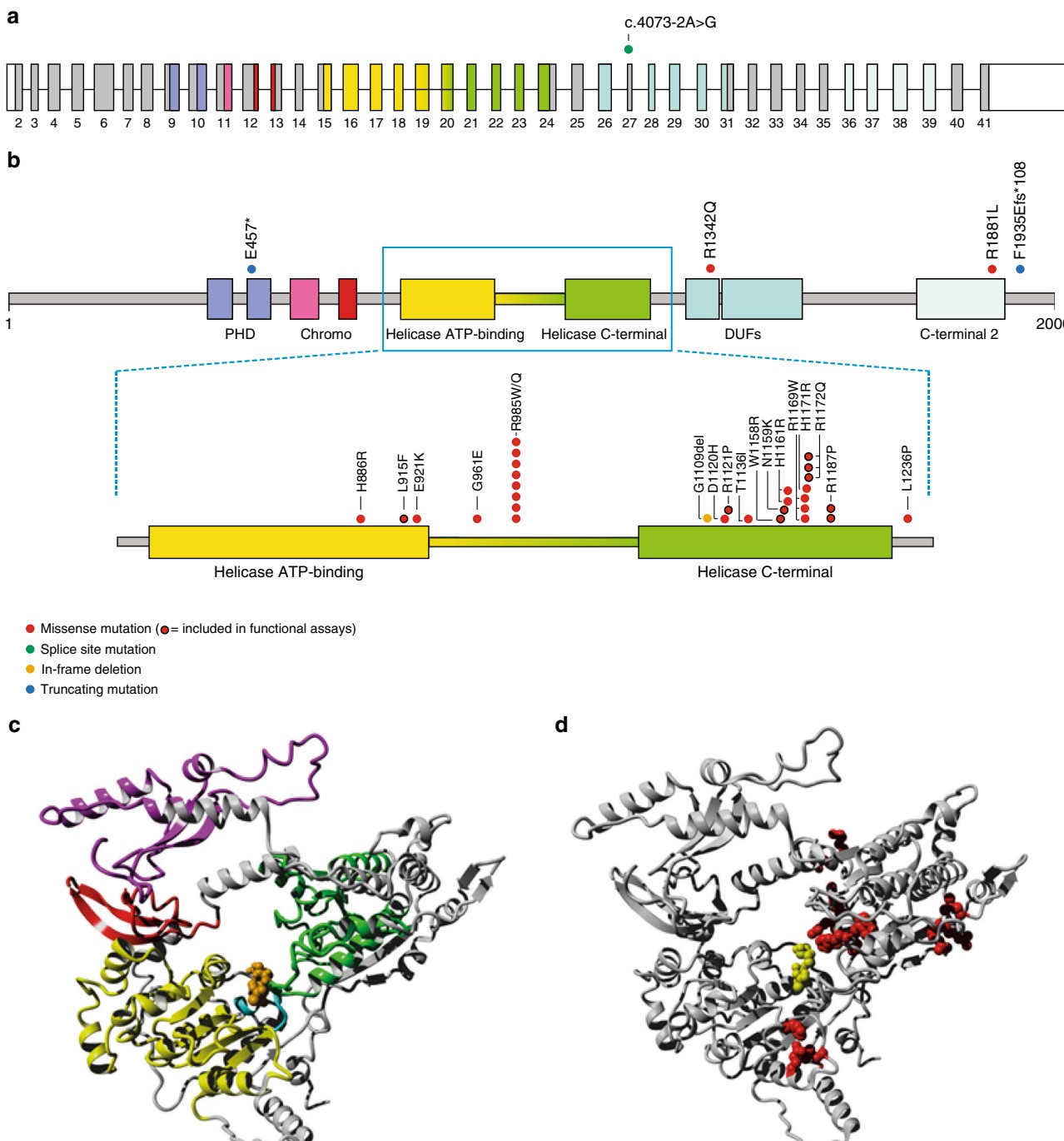

**Fig. 2** Schematic view of CHD3 transcript and protein with de novo mutations. **a** Schematic view of CHD3 exons (transcript 1, NM_001005273.2) with the splice site mutation c.4073-2A>G shown that most likely leads to skipping of exon 27 (22 amino acids), while preserving the reading frame. Exon 27 is part of the beginning of the second DUF domain (DUF 1086). Colors of the domains in **a** match with colors of domains in **b** and **c**. Five different types of domains are specified: plant homeodomains (PHD), chromodomains (Chromo), a Helicase domain consisting of two parts (Helicase ATP-binding and Helicase C-terminal), domains of unknown function (DUF), and a C-terminal 2 domain. **b** Schematic view of linear CHD3 protein (transcript 1, NM_001005273.2) with all mutations, except for the splice site mutation that is shown in **a**, found in our cohort. Almost all missense mutations cluster in or around the Helicase domain of the CHD3 protein. **c** Overview of one of the two CHD3-models used in this study, based on the 3MWY protein structure. This figure shows the different domains of the protein in their three-dimensional conformation: chromo domain 1 494–595 (magenta), chromo domain 2 631–673 (red), helicase ATP binding domain (yellow), helicase C-terminal domain (green), ATP binding residues 761–768 (cyan). ATP is orange, and gray residues do not belong to an indicated domain. Colors of the domains in **c** match with colors of domains in **a** and **b**. **d** The same structure as **c**, but in this figure the positions of the mutated residues are indicated in red, the sidechains of these residues are shown as red balls. The ATP molecule is shown in yellow. This figure illustrates the clustering of mutations on specific sites within the Helicase ATP-binding domain and Helicase C-terminal domain. A more detailed analysis of the different missense mutations in our cohort can be found in Supplementary Note 1

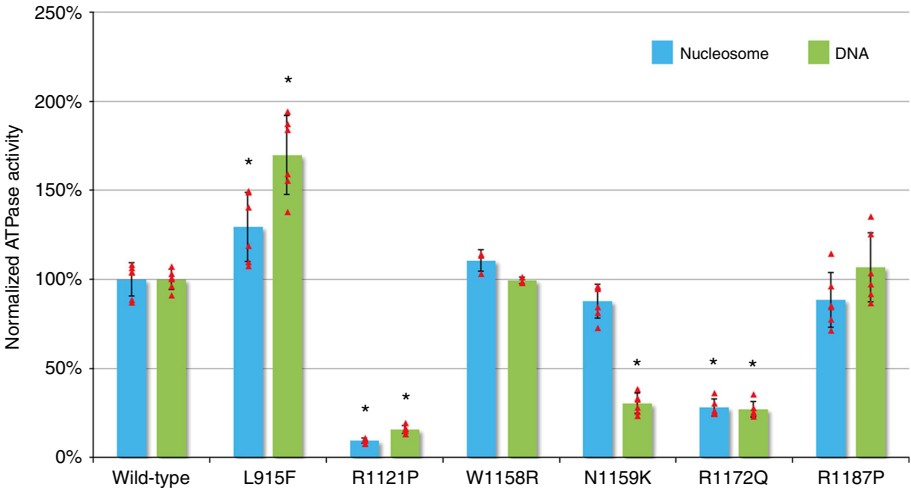

**Fig. 3** ATPase assays. Radiometric ATPase assays were performed to assess the activity of the mutant proteins relative to wild-type, in the presence of recombinant nucleosomes (blue), dsDNA (green), or in the absence of DNA substrates as a control (Supplementary Fig. 4). Released phosphate was separated from unhydrolyzed ATP by thin layer chromatography, and detected by exposure to a phosphorimager. The experimental values (percentage hydrolyzed ATP) for the different mutant conditions were normalized to values for the wild-type condition within the experiment, to derive a normalized ATPase activity. The experimental data are presented as means ± standard deviation, individual data points are shown as red triangles. Three independent experiments from two individual purifications (wild-type, p.Leu915Phe, p.Arg1121Pro, p.Asn1159Lys, p.Arg1172Gln, and p.Arg1187Pro) ($N = 6$) or one purification (p.Trp1158Arg) ($N = 3$) were performed. Raw values from the individual experiments can be found in Supplementary Data 2. Asterisk (*) indicates significant difference for mutant values compared to wild-type values (unpaired $t$-test, $P < 0.05$) within the same substrate condition

chromatin, with some mutants exhibiting decreased activity while one shows increased activity.

## Discussion

In this study, we show that de novo *CHD3* mutations cause a neurodevelopmental disorder. We demonstrate defining clinical features of this syndrome. The characteristic phenotype of individuals with *CHD3* mutations overlaps with that reported for de novo mutations in *CHD4*, in which intellectual disability, macrocephaly, ventriculomegaly, undescended testes, and similar facial features have been reported. However, comparisons to the CHD4-related syndrome are currently limited because so far only five individuals with *CHD4* mutations have been clinically characterized. Also interesting in this context is the fact that four of the six recently described patients with missense mutations in *CHD1* have a diagnosis of speech apraxia[9], a relatively rare condition. Although CHD1 does not function in the same protein complex as CHD3 and has different expression patterns[9], there might be shared pathogenic mechanisms leading to speech problems in patients with mutations in these chromatin remodelers.

Based on the molecular and phenotypic data of individuals in our cohort, there is no obvious correlation between the precise type or location of the mutation, and the severity of the variable features of the resulting syndrome. However, the only individual in our cohort with epilepsy is also the only case with a missense mutation in the C-terminal domain of the protein. Future identification of more individuals with missense mutations in this region of the protein will help resolve whether this reflects a phenotype–genotype correlation.

In addition to defining the phenotype associated with *CHD3* mutations, we aimed to characterize the effects of *CHD3* mutations at a molecular and functional level. ATPase assays with six different mutant CHD3 constructs showed a clearly decreased ATPase activity for two mutations (p.Arg1121Pro and p.Arg1172Gln) and increased ATPase activity for one mutation (p.Leu915Phe). The disturbed ATPase activities are associated with corresponding effects on chromatin remodeling capacities for these three mutants,

as shown by the restriction enzyme accessibility assays. It is currently unclear how deactivating and activating mutations can both yield similarly disruptive effects on neurodevelopmental outcomes. However, a recent study of cancer-specific mutations in the chromatin remodeling ATPase SMARCA4 concluded that mutations in the ATPase core of this enzyme had dominant-negative impacts on the global chromatin landscape regardless of whether they displayed increased or decreased dynamic recovery in fluorescence after photobleaching[28]. By analogy, it seems plausible that perturbed chromatin remodeling activity of CHD3, whether by gain or loss of activity relative to wild-type or by affecting its interactions, might likewise alter chromatin landscapes, to contribute to a neurodevelopmental phenotype.

Two mutations (p.Trp1158Arg and p.Asn1159Lys) show severely decreased chromatin remodeling capacities, despite unaffected ATPase activity in the presence of recombinant nucleosomes. In line with these findings, the highly conserved tryptophan residue at a position analogous to CHD3 residue 1158 has recently been shown to be critical for chromatin remodeling, but not for ATP hydrolysis, in the context of yeast SNF2[27]. Interestingly, the mutation in our cohort affecting this amino acid (p.Trp1158Arg) directly matches the position of a previously published mutation in CHD4[8] (Supplementary Fig. 1), while the other previously published de novo missense mutations in CHD4-related syndrome are also mainly affecting the orthologous Helicase domain of CHD4 (Supplementary Fig. 1)[8,29].

To systematically assess whether the distribution of the missense mutations in CHD3 reflects mutational hotspots in the gene, we performed a formal clustering analysis based on mutual distances, as previously described[30]. This analysis revealed significant clustering within the transcript ($P = 0.0017$), a finding that argues against simple haploinsufficiency as an underlying molecular mechanism. The paucity of patients with truncating mutations compared to the 31 patients with missense mutations in our cohort also supports this view, although the precise mechanistic effects of CHD3 mutations during neurodevelopment are a topic for future study.

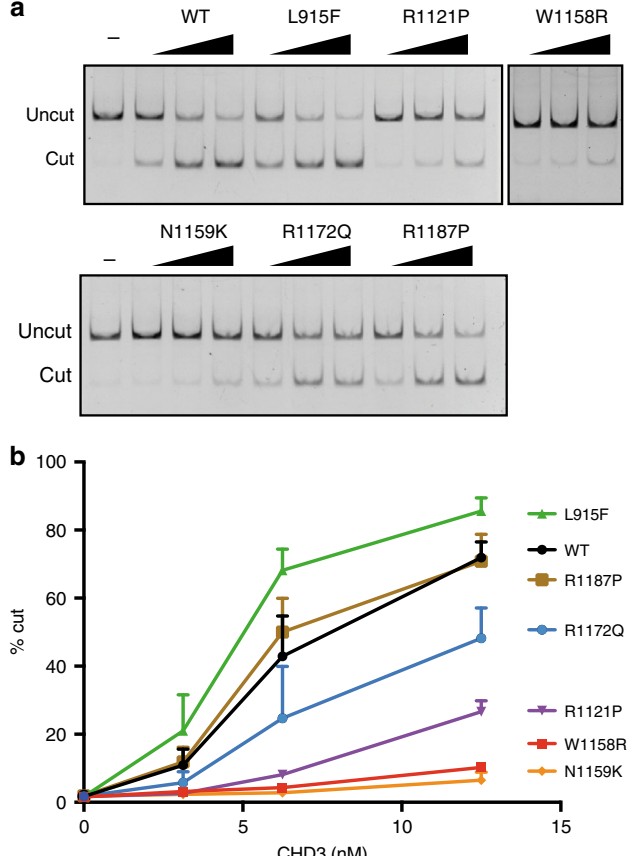

**Fig. 4** Restriction enzyme accessibility assay. **a** Restriction enzyme accessibility analysis of CHD3 wild-type and mutant proteins. 3.125, 6.25, or 12.5 nM of CHD3 proteins were incubated with 347 bp mononucleosomes. Digested fragments were analyzed by native polyacrylamide gel. **b** Quantitative analysis of restriction enzyme accessibility. Three individual experiments from two individual purifications (wild-type, p.Leu915Phe, p.Arg1121Pro, p.Asn1159Lys, p.Arg1172Gln, and p.Arg1187Pro) ($N = 6$) or one purification (p.Trp1158Arg) ($N = 3$) were conducted. The experimental data are presented as means with standard deviations

Taken together, with our research we identify a recognizable neurodevelopmental disorder. We define the phenotypic spectrum associated with mutations in *CHD3*, and show the effects of several different mutations on ATPase activity and chromatin remodeling capacities. Our findings highlight the importance of chromatin remodeling factors, and specifically the CHD3 protein, in human brain development.

## Methods

**Individuals and consents**. The authors affirm that (the legal representatives of) all human research participants provided informed consent for publication of the images in Fig. 1. Informed consent was also derived for the use of biological materials from all individuals or their legal representative. Genetic testing and research were performed in accordance with protocols approved by the local Institutional Review Boards where the patients were followed. Specifically, research exomes were performed after informed consent on protocols approved by the Institutional Review Boards of the following institutions: University of British Columbia, Augustana College, CHU Dijon, Mass General Hospital for Children, University of Erlangen-Nuremberg, Hamburg Chamber of Physicians, Cambridge South—UK Research Ethics Committee, University of Wisconsin-Madison Social & Behavioral Sciences.

**Annotation of mutations**. All mutations in this report are annotated in GRCh37 (hg19) and CHD3 transcript variant 1 (NM_001005273.2).

**Next-generation sequencing**. For the index case (individual 22), whole genome sequencing was performed using Illumina's HiSeq X Ten technology, the Burrows–Wheeler Aligner (BWA) software version 0.7.8-r455[31] and GATK v.3.4[32]. In other individuals, exome or genome sequencing and data analysis were performed as previously described[33–44].

**Expression and purification of FLAG-CHD3**. CHD3 proteins were prepared as previously described[45], with the following modifications. FLAG-CHD3 constructs were cloned into expression vectors (kindly provided by Guang Hu) using Gateway Cloning technology. Primer sequences are provided in Supplementary Table 1. HEK293-f (ThermoFisher, FreeStyle™ 293-F Cells) were grown in suspension culture using FreeStyle™ 293 Expression Medium (ThermoFisher) in optimum growth flasks (Thomson) using a shaking incubator set at 8% $CO_2$, 80% humidity, and 150 rpm shaker rate. The cell count was $10^6$ cells/ml on the day of transfection. Cells were transfected with 1 mg of expression vector using PEI max (Polysciences). Cells were harvested 48 h after transfection by centrifugation at $400 \times g$ for 6 min. Cells were washed once with phosphate buffer saline solution prior to storage at −80 °C or protein purification.

The cell pellet was resuspended in lysis buffer (20 mM HEPES, 1.5 mM $MgCl_2$, 10 mM KCl, 1 mM DTT, 1 mM PMSF, and 1× cOmplete® protease-inhibitor EDTA-free (Roche), pH 7.6). Cells were incubated on ice for 30 min, vortexed briefly, and nuclei were collected by centrifugation (5 min, $3300 \times g$, 4 °C). The supernatant was discarded and the nuclear pellet was resuspended in nuclear extraction buffer (20 mM HEPES, 0.5 M KCl, 1.5 mM $MgCl_2$, 0.2 mM EDTA, 20% glycerol, 0.2% NP-40, 1 mM DTT, 1 mM PMSF, and 1× cOmplete® protease-inhibitor EDTA-free (Roche), pH 7.6). The nuclear pellet was homogenized using a Dounce homogenizer, incubated on ice for 30 min, and insoluble material was removed by centrifugation (20 min, $110,000 \times g$, 4 °C). The supernatant (nuclear extract) was incubated with α-FLAG M2 affinity gel (Sigma-Aldrich) and rotated overnight at 4 °C. The α-FLAG beads were then washed twice with nuclear extraction buffer, followed by 2 additional washes with wash buffer (20 mM HEPES, 0.1 M KCl, 0.2% NP-40, 20% glycerol, and 1 mM DTT, pH 7.6). The FLAG-CHD3 protein was eluted with 0.3 mg/ml 3XFLAG peptide (in 20 mM HEPES, 0.1 M KCl, 0.05% NP-40, 20% glycerol, and 1 mM DTT, pH 7.6). Wild-type and mutant protein samples were analyzed by SDS-PAGE and stained with Coomassie Brilliant Blue (Supplementary Fig. 4). The concentration of the CHD3 proteins was estimated from BSA standards in SDS-PAGE gels stained with Coomassie Brilliant Blue.

**Radiometric ATPase assay**. Each ATPase reaction (10 µL) contained 20 mM Tris–HCl, pH 7.5, 1 mM $MgCl_2$, 0.1 mg/ml BSA, 1 mM DTT, 100 µM ATP, 1 µCi of [γ-$^{32}$P]ATP as a tracer. 25 nM of each CHD3 purified protein was incubated with 70 nM of recombinant nucleosomes or naked dsDNA. Nucleosome was reconstituted by the salt gradient dialysis method using recombinant histone octamer and 201 bp 601 DNA fragment[46]. The reactions were initiated by the addition of nucleosome or DNA substrate and incubated at 37 °C for 40 min. The reaction was quenched by the addition of EDTA to a final concentration of 100 mM. Aliquots (2.5 µL) were removed and spotted on PEI-cellulose thin-layer chromatography plates and developed in 1 M formic acid and 0.5 M LiCl. ATP hydrolysis was quantified using a Phosphorimager with Image Quant Software. For the mixing experiment, all reaction components except for CHD3 protein were incubated for 10 min at 37 °C, and the CHD3 protein mixture was added last to start the reaction. This experiment was performed 3 times per condition ($N = 3$) for all conditions, except for the conditions "no CHD3", "WT 12.5 nM" and "WT 25 nM" ($N = 2$).

For the quantification analysis, we performed 3 individual experiments for each of the two biological replicates (total $N = 6$), except for the p.Trp1158Arg mutant (one biological replicate, total $N = 3$). An unpaired *t*-test was used to determine whether the activity of the mutant proteins differed significantly from wild-type protein activity.

**Restriction enzyme accessibility assay**. Remodeling activities were measured with a restriction enzyme accessibility assay as previously described[27]. 12.5 nM nucleosomes (347 bp) were incubated with the indicated amounts of CHD3 proteins at 37 °C for 60 min in the remodeling buffer (20 mM Tris–HCl pH 7.5, 1 mM DTT, 1 mM $MgCl_2$, 1 mM ATP, 0.1 mg/ml BSA, and 5 U HhaI). The reactions were stopped by adding 2 µL of proteinase K buffer containing 6.7 mg/ml proteinase K, 167 mM EDTA, and 1.7% SDS. After incubation at 50 °C for 10 min, the DNAs were analyzed by 6% native polyacrylamide gel electrophoresis. The separated DNA fragments were visualized with UV light on the ChemiDox XRS system (BIO-RAD). The band intensities were quantified by ImageJ.

**Cloning constructs for immunofluorescence**. Wild-type CHD3 (NM_001005273.2) was amplified by PCR and cloned into pCR2.1-TOPO (Invitrogen) as described[47]. CHD3 mutation constructs were generated using the QuikChange II Site-Directed Mutagenesis Kit (Agilent), primer sequences are provided in Supplementary Table 1. CHD3 cDNAs were subcloned using BamHI/NheI restriction sites into a modified pmCherry-C1 vector (Clontech). All constructs were verified by Sanger sequencing.

**Immunofluorescence**. HEK293 cells were obtained from ECACC (Catalogue number 85120602) and grown in Dulbecco's modified Eagle's medium (Invitrogen), supplemented with 10% fetal bovine serum (Invitrogen). Transfection was performed using GeneJuice (Merck-Millipore). The cells were seeded onto coverslips coated with poly-L-lysine (Sigma). At 36 h post-transfection, cells were fixed using 4% paraformaldehyde solution (Electron Microscopy Sciences) for 10 min at room temperature. The mCherry fusion proteins were visualized by direct fluorescence, nuclei were visualized with Hoechst 33342 (Invitrogen). Fluorescence images were obtained using an Axiovert A-1 fluorescent microscope with ZEN Image Software (Zeiss).

**Three-dimensional modeling**. As no experimentally solved 3D-structure of CHD3 exists, we performed homology modeling using the modeling option with standard parameters in the YASARA[48] & WHAT IF[49] twinset. Several models of the ATPase/helicase domain were created. The best scoring model was based on template PDB-file 5JXR (sequence identity 41% over the aligned residues). We also studied the model based on PDB-file 3MWY (sequence identity 45%), which shows a more open conformation and contains an ATP substitute. These two models provided information about the relative position of the mutated residues in the different conformation of the protein complex.

**Clustering analysis of missense mutations**. The locations of observed de novo missense mutations were permutated 1,000,000 times over the cDNA of the *CHD3* gene (RefSeq transcript: NM_001005273.2). The distances between missense mutations were adjusted to take into account the total size of the coding region of *CHD3* (6003 bp). Then, the geometric mean (the *n*th root of the product of *n* of all distances separating the mutations) was calculated, giving an index of clustering, as previously described[30]. To circumvent a mean distance of 0 as the result of recurrent mutations, pseudocount (adding 1 to all distances and 1 to the gene size) was used. To avoid artificial deflation of the clustering *P*-value, only one of the recurrent mutations present in the sibling-pair (individuals 7 and 8) and twin-pair (individuals 20 and 21) were included for the analysis.

## Data availability

All genotypic and phenotypic data supporting the findings of this study are available within the paper and supplementary files. Data are also freely available in the ClinVar database, under accession numbers SCV000787629–SCV000787651. Raw data of functional experiments are available from the corresponding authors (P.M.C. and S.E.F.) upon request.

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

## Acknowledgements

We thank all individuals and families for their contribution. We thank Amaia Carrión Castillo and Else Eising for assistance with the WGS analysis of the index individual, and Sarah Graham and Elliot Sollis for cloning the wild-type CHD3 construct for immunofluorescence. This work was supported by the Netherlands Organization for Scientific Research (NWO) Gravitation Grant 24.001.006 to the Language in Interaction Consortium (L.S.B., S.E.F., and H.G.B.), the Max Planck Society (S.E.F.), the National Institute on Deafness and Other Communication Disorders Grant DC000496 (L.Sh.) and a core grant to the Waisman Center from the National Institute of Child Health and Human Development (Grant U54 HD090256) to L.Sh., the Canadian Institutes of Health Research Grants MOP-119595 and PJT-148830 to W.T.G. Individuals 11, 16, 24, and 28 were part of The DDD Study cohort. The DDD Study presents independent research commissioned by the Health Innovation Challenge Fund [Grant number HICF-1009-003], a parallel funding partnership between the Wellcome Trust and the Department of Health, and the Wellcome Trust Sanger Institute [Grant number WT098051]. The views expressed in this publication are those of the author(s) and not necessarily those of the Wellcome Trust or the Department of Health. The DDD study has UK Research Ethics Committee approval (10/H0305/83, granted by the Cambridge South REC, and GEN/284/12, granted by the Republic of Ireland REC). The research team acknowledges the support of the National Institute for Health Research, through the Comprehensive Clinical Research Network.

## Author contributions

L.S.B., S.E.F., P.A.W. and P.M.C. designed the study. J.Ro., J.T., S.E., M.T., J.D.R., R.M.P., P.A.W. and P.M.C. were involved in the design and execution of the ATPase assays. L.S.B. performed the immunofluorescence work with supervision of P.D. The three-dimensional modeling was performed by H.V. and R.P. performed the RNA analysis of the frameshift mutation. L.S.B., L.H.R., C.B.N., J.D., K.J.S., M.A.S., I.S., C.T.R.M.S., A.P.A.S., P.W., M.W., E.F., A.K., W.T.G., A.S.A.C., R.A., A.M.I., P.Y.B.A., J.Ra., I.J.A., S.A.S., R.J.L., H.E.W., A.A., B.K., C.N., J.B., A.I., D.R., R.L., J.P., T.E., M.C., S.L., J.H.C., S.P., R.E.S., G.D., I.M.W., C.Z., A.R., M.G.B., C.M., M.K., E.H.B., G.R.M., K.L.I.G., E.B., R.N.E., L.B., I.B., H.M., S.B.W., K.J.J., E.A.S., K.K., T.B., S.M., H.K., S.L., R.P., S.J., L.F., J.T., M.A., L.S., T.K., H.B. and P.M.C. participated in recruitment of individuals, phenotyping and/or next-generation sequencing analysis. L.S.B., J.Ro., J.T., S.E., M.T., J.D.R., R.M.P., T.K., H.G.B., P.A.W., S.E.F. and P.M.C. analyzed and interpreted the results and wrote the manuscript. T.K., H.G.B., P.A.W., P.M.C. and S.E.F. supervised the project. All authors contributed to the final version of the manuscript.

## Additional information

**Competing interests:** The authors declare no competing interests.

Lot Snijders Blok [1,2,3], Justine Rousseau[4], Joanna Twist[5], Sophie Ehresmann[4], Motoki Takaku[5], Hanka Venselaar[6], Lance H. Rodan[7], Catherine B. Nowak[7], Jessica Douglas[7], Kathryn J. Swoboda[8], Marcie A. Steeves[9], Inderneel Sahai[9], Connie T.R.M. Stumpel[10], Alexander P.A. Stegmann[10], Patricia Wheeler[11], Marcia Willing[12], Elise Fiala[12], Aaina Kochhar[13], William T. Gibson[14,15], Ana S.A. Cohen [14,15], Ruky Agbahovbe[14,15], A. Micheil Innes [16], P.Y.Billie Au[16], Julia Rankin[17], Ilse J. Anderson[18], Steven A. Skinner[19], Raymond J. Louie[19], Hannah E. Warren[19], Alexandra Afenjar[20], Boris Keren[21,22], Caroline Nava[21,22,23], Julien Buratti[21], Arnaud Isapof[24], Diana Rodriguez[25], Raymond Lewandowski[26], Jennifer Propst[26], Ton van Essen[27], Murim Choi[28], Sangmoon Lee[28], Jong H. Chae[29], Susan Price[30], Rhonda E. Schnur[31], Ganka Douglas[31], Ingrid M. Wentzensen[31], Christiane Zweier[32], André Reis[32], Martin G. Bialer[33], Christine Moore[33], Marije Koopmans[34], Eva H. Brilstra[34], Glen R. Monroe[34], Koen L.I. van Gassen[34], Ellen van Binsbergen[34], Ruth Newbury-Ecob[35], Lucy Bownass[35], Ingrid Bader[36], Johannes A. Mayr[37], Saskia B. Wortmann[37,38,39], Kathy J. Jakielski[40], Edythe A. Strand[41], Katja Kloth[42], Tatjana Bierhals[42], The DDD study, John D. Roberts[5], Robert M. Petrovich[5], Shinichi Machida[43], Hitoshi Kurumizaka [43], Stefan Lelieveld[1], Rolph Pfundt[1], Sandra Jansen[1,3], Pelagia Deriziotis[2],

Laurence Faivre[44,45], Julien Thevenon[44,45], Mirna Assoum[44,45], Lawrence Shriberg[46], Tjitske Kleefstra[1,3], Han G. Brunner[1,3,10], Paul A. Wade [5], Simon E. Fisher [2,3] & Philippe M. Campeau [4,47]

[1]Department of Human Genetics, Radboud University Medical Center, Nijmegen 6500HB, The Netherlands. [2]Language and Genetics Department, Max Planck Institute for Psycholinguistics, Nijmegen 6500AH, The Netherlands. [3]Donders Institute for Brain, Cognition and Behaviour, Radboud University, Nijmegen 6500HE, The Netherlands. [4]CHU Sainte-Justine Research Center, Montreal QC H3T 1C5, Canada. [5]National Institute of Environmental Health Sciences, Research Triangle Park NC 27709, USA. [6]Centre for Molecular and Biomolecular Informatics, Radboud Institute for Molecular Life Sciences, Radboud University Medical Center, Nijmegen 6500HB, The Netherlands. [7]Division of Genetics and Genomics, Boston Children's Hospital, Boston MA 02115, USA. [8]Department of Neurology, Massachusetts General Hospital and Harvard Medical School, Boston MA 02114, USA. [9]Department of Medical Genetics, Massachusetts General Hospital, Boston MA 02114, USA. [10]Department of Clinical Genetics and GROW-School for Oncology and Developmental Biology, Maastricht University Medical Center, Maastricht 6202AZ, The Netherlands. [11]Nemours Childrens Clinic, Orlando FL 32827, USA. [12]Division of Genetics and Genomic Medicine, Department of Pediatrics, Washington University School of Medicine, St. Louis MO 63110, USA. [13]Valley Children's Hospital, Madera CA 93636, USA. [14]British Columbia Children's Hospital Research Institute, Vancouver BC V5Z 4H4, Canada. [15]Department of Medical Genetics, University of British Columbia, Vancouver BC V6H 3N1, Canada. [16]Department of Medical Genetics and Alberta Children's Hospital Research Institute, Cumming School of Medicine, University of Calgary, Calgary AB T2N 4N1, Canada. [17]Department of Clinical Genetics, Royal Devon and Exeter NHS Foundation Trust (Heavitree), Exeter EX2 5DW, UK. [18]Division of Genetics, Department of Medicine, University of Tennessee Medical Center, Knoxville TN 37920, USA. [19]Greenwood Genetic Center, Greenwood SC 29646, USA. [20]GRC ConCer-LD, Sorbonne Universités, UPMC Univ Paris ; Department of Medical Genetics and Centre de Référence Malformations et maladies congénitales du cervelet et déficiences intellectuelles de causes rares, Armand Trousseau Hospital, GHUEP, AP-HP, Paris 75012, France. [21]AP-HP, Hôpital de la Pitié-Salpêtrière, Département de Génétique, Paris 75013, France. [22]Groupe de Recherche Clinique (GRC) 'déficience intellectuelle et autisme' UPMC, Paris 75005, France. [23]INSERM, U 1127, CNRS UMR 7225, Institut du Cerveau et de la Moelle épinière, ICM, Sorbonne Universités, UPMC Univ Paris 06 UMR S 1127, 75013 Paris, France. [24]GRC ConCer-LD, Sorbonne Universités, UPMC Univ Paris 06; Department Child Neurology and Reference Center for Neuromuscular Diseases "Nord/Est/Ile-de-France", FILNEMUS, Armand Trousseau Hospital, GHUEP, AP-HP, Paris 75012, France. [25]GRC ConCer-LD, Sorbonne Universités, UPMC Univ Paris 06; Department of Child Neurology and National Reference Center for Neurogenetic Disorders, Armand Trousseau Hospital, GHUEP, AP-HP, INSERM U1141, 75012 Paris, France. [26]Clinical Genetics Division, Virginia Commonwealth University Health System, Richmond VA 23298, USA. [27]Clinical Genetics Department, University Medical Center Groningen, Groningen 9700RB, The Netherlands. [28]Department of Biomedical Sciences, Seoul National University College of Medicine, Seoul 08826, Republic of Korea. [29]Department of Pediatrics, Seoul National University College of Medicine, Seoul National University Children's Hospital, Seoul 08826, Republic of Korea. [30]Oxford University Hospitals NHS Foundation Trust, Oxford OX3 7HE, UK. [31]GeneDx, Gaithersburg MD 20877, USA. [32]Institute of Human Genetics, Friedrich-Alexander-Universität Erlangen-Nürnberg, Erlangen 91054, Germany. [33]Northwell Health, Division of Medical Genetics and Genomics, Great Neck NY 11021, USA. [34]Department of Genetics, University Medical Center Utrecht, Utrecht University, Utrecht 3508AB, The Netherlands. [35]University Hospitals Bristol, Department of Clinical Genetics, St Michael's Hospital, Bristol BS2 8EG, UK. [36]Department of Clinical Genetics, University Children's Hospital, Paracelsus Medical University, Salzburg A-5020, Austria. [37]Department of Pediatrics, Salzburger Landeskliniken and Paracelsus Medical University, Salzburg A-5020, Austria. [38]Institute of Human Genetics, Technische Universität München, Munich 81675, Germany. [39]Institute of Human Genetics, Helmholtz Zentrum München, Neuherberg 85764, Germany. [40]Communication Sciences and Disorders, Augustana College, Rock Island IL 61201, USA. [41]Department of Neurology, Mayo Clinic, Rochester MN 55905, USA. [42]Institute of Human Genetics, University Medical Center Hamburg-Eppendorf, Hamburg 20246, Germany. [43]Waseda University, Tokyo 169-8050, Japan. [44]Equipe Génétique des Anomalies du Développement, Université de Bourgogne-Franche Comté, Dijon 21070, France. [45]Centre de Génétique et Centre de Référence Anomalies du Développement et Syndromes Malformatifs, FHU TRANSLAD, Hôpital d'Enfants, CHU Dijon et Université de Bourgogne, Dijon 21079, France. [46]Waisman Center, Phonology Project, Madison WI 53705-2280, USA. [47]Sainte-Justine Hospital, University of Montreal, Montreal QC H3T 1C5, Canada. These authors contributed equally: Lot Snijders Blok, Justine Rousseau, Joanna Twist. These authors jointly supervised this work: Simon E. Fisher, Philippe M. Campeau. A full list of consortium members appears below.

## The DDD study

Jeremy F. McRae[48], Stephen Clayton[48], Tomas W. Fitzgerald[48], Joanna Kaplanis[48], Elena Prigmore[48], Diana Rajan[48], Alejandro Sifrim[48], Stuart Aitken[49], Nadia Akawi[48], Mohsan Alvi[50], Kirsty Ambridge[48], Daniel M. Barrett[48], Tanya Bayzetinova[48], Philip Jones[48], Wendy D. Jones[48], Daniel King[48], Netravathi Krishnappa[48], Laura E. Mason[48], Tarjinder Singh[48], Adrian R. Tivey[48], Munaza Ahmed[51,52,53], Uruj Anjum[54], Hayley Archer[55,56], Ruth Armstrong[57], Jana Awada[48], Meena Balasubramanian[58], Siddharth Banka[59], Diana Baralle[51,52,53], Angela Barnicoat[60], Paul Batstone[61], David Baty[62], Chris Bennett[63], Jonathan Berg[62], Birgitta Bernhard[64], A. Paul Bevan[48], Maria Bitner-Glindzicz[60], Edward Blair[65], Moira Blyth[63], David Bohanna[66], Louise Bourdon[64], David Bourn[67], Lisa Bradley[68], Angela Brady[64], Simon Brent[48], Carole Brewer[69], Kate Brunstrom[60], David J. Bunyan[51,52,53], John Burn[67], Natalie Canham[64], Bruce Castle[69], Kate Chandler[59], Elena Chatzimichali[48], Deirdre Cilliers[65], Angus Clarke[55,56], Susan Clasper[65], Jill Clayton-Smith[59], Virginia Clowes[64], Andrea Coates[63], Trevor Cole[66], Irina Colgiu[48], Amanda Collins[51,52,53], Morag N. Collinson[51,52,53], Fiona Connell[70], Nicola Cooper[66], Helen Cox[66], Lara Cresswell[71], Gareth Cross[72],

Yanick Crow[59], Mariella D'Alessandro[61], Tabib Dabir[68], Rosemarie Davidson[73], Sally Davies[55,56], Dylan de Vries[48], John Dean[61], Charu Deshpande[70], Gemma Devlin[69], Abhijit Dixit[72], Angus Dobbie[63], Alan Donaldson[74], Dian Donnai[59], Deirdre Donnelly[68], Carina Donnelly[59], Angela Douglas[75], Sofia Douzgou[59], Alexis Duncan[73], Jacqueline Eason[72], Sian Ellard[69], Ian Ellis[75], Frances Elmslie[54], Karenza Evans[55,56], Sarah Everest[69], Tina Fendick[70], Richard Fisher[67], Frances Flinter[70], Nicola Foulds[51,52,53], Andrew Fry[55,56], Alan Fryer[75], Carol Gardiner[73], Lorraine Gaunt[59], Neeti Ghali[64], Richard Gibbons[65], Harinder Gill[76], Judith Goodship[67], David Goudie[62], Emma Gray[48], Andrew Green[76], Philip Greene[48], Lynn Greenhalgh[75], Susan Gribble[48], Rachel Harrison[72], Lucy Harrison[51,52,53], Victoria Harrison[51,52,53], Rose Hawkins[74], Liu He[48], Stephen Hellens[67], Alex Henderson[67], Sarah Hewitt[63], Lucy Hildyard[48], Emma Hobson[63], Simon Holden[57], Muriel Holder[64], Susan Holder[64], Georgina Hollingsworth[60], Tessa Homfray[54], Mervyn Humphreys[68], Jane Hurst[60], Ben Hutton[48], Stuart Ingram[58], Melita Irving[70], Lily Islam[66], Andrew Jackson[49], Joanna Jarvis[66], Lucy Jenkins[60], Diana Johnson[58], Elizabeth Jones[59], Dragana Josifova[70], Shelagh Joss[73], Beckie Kaemba[71], Sandra Kazembe[71], Rosemary Kelsell[48], Bronwyn Kerr[59], Helen Kingston[59], Usha Kini[65], Esther Kinning[73], Gail Kirby[66], Claire Kirk[68], Emma Kivuva[69], Alison Kraus[63], Dhavendra Kumar[55,56], V.K.Ajith Kumar[60], Katherine Lachlan[51,52,53], Wayne Lam[49], Anne Lampe[49], Caroline Langman[70], Melissa Lees[60], Derek Lim[66], Cheryl Longman[73], Gordon Lowther[73], Sally A. Lynch[76], Alex Magee[68], Eddy Maher[49], Alison Male[60], Sahar Mansour[54], Karen Marks[54], Katherine Martin[72], Una Maye[75], Emma McCann[77], Vivienne McConnell[68], Meriel McEntagart[54], Ruth McGowan[61], Kirsten McKay[66], Shane McKee[68], Dominic J. McMullan[66], Susan McNerlan[68], Catherine McWilliam[61], Sarju Mehta[57], Kay Metcalfe[59], Anna Middleton[48], Zosia Miedzybrodzka[61], Emma Miles[59], Shehla Mohammed[70], Tara Montgomery[67], David Moore[49], Sian Morgan[55,56], Jenny Morton[66], Hood Mugalaasi[55,56], Victoria Murday[73], Helen Murphy[59], Swati Naik[66], Andrea Nemeth[65], Louise Nevitt[58], Andrew Norman[66], Rosie O'Shea[76], Caroline Ogilvie[70], Kai-Ren Ong[66], Soo-Mi Park[57], Michael J. Parker[58], Chirag Patel[66], Joan Paterson[57], Stewart Payne[64], Daniel Perrett[48], Julie Phipps[65], Daniela T. Pilz[73], Martin Pollard[48], Caroline Pottinger[77], Joanna Poulton[65], Norman Pratt[62], Katrina Prescott[63], Abigail Pridham[65], Annie Procter[55,56], Hellen Purnell[65], Oliver Quarrell[58], Nicola Ragge[66], Raheleh Rahbari[48], Josh Randall[48], Lucy Raymond[57], Debbie Rice[62], Leema Robert[70], Eileen Roberts[74], Jonathan Roberts[57], Paul Roberts[63], Gillian Roberts[75], Alison Ross[61], Elisabeth Rosser[60], Anand Saggar[54], Shalaka Samant[61], Julian Sampson[55,56], Richard Sandford[57], Ajoy Sarkar[72], Susann Schweiger[62], Richard Scott[60], Ingrid Scurr[74], Ann Selby[72], Anneke Seller[65], Cheryl Sequeira[64], Nora Shannon[72], Saba Sharif[66], Charles Shaw-Smith[69], Emma Shearing[58], Debbie Shears[65], Eamonn Sheridan[63], Ingrid Simonic[57], Roldan Singzon[64], Zara Skitt[59], Audrey Smith[63], Kath Smith[58], Sarah Smithson[74], Linda Sneddon[67], Miranda Splitt[67], Miranda Squires[63], Fiona Stewart[68], Helen Stewart[65], Volker Straub[67], Mohnish Suri[72], Vivienne Sutton[75], Ganesh Jawahar Swaminathan[48], Elizabeth Sweeney[75], Kate Tatton-Brown[54], Cat Taylor[5], Rohan Taylor[54], Mark Tein[66], I. Karen Temple[51,52,53], Jenny Thomson[63], Marc Tischkowitz[57], Susan Tomkins[74], Audrey Torokwa[51,52,53], Becky Treacy[57], Claire Turner[69], Peter Turnpenny[69], Carolyn Tysoe[69], Anthony Vandersteen[64], Vinod Varghese[55,56], Pradeep Vasudevan[71], Parthiban Vijayarangakannan[48], Julie Vogt[66], Emma Wakeling[64], Sarah Wallwark[57], Jonathon Waters[60], Astrid Weber[75], Diana Wellesley[51,52,53], Margo Whiteford[73], Sara Widaa[48], Sarah Wilcox[57], Emily Wilkinson[48], Denise Williams[66], Nicola Williams[73], Louise Wilson[60], Geoff Woods[57], Christopher Wragg[74], Michael Wright[67], Laura Yates[67], Michael Yau[70], Chris Nellåker[78,79,80], Michael Parker[81], Helen V. Firth[48,57], Caroline F. Wright[48], David R. FitzPatrick[48,49], Jeffrey C. Barrett[48] & Matthew E. Hurles[48]

[48]Wellcome Trust Sanger Institute, Wellcome Trust Genome Campus, Hinxton, Cambridge CB10 1SA, UK. [49]MRC Human Genetics Unit, MRC IGMM, University of Edinburgh, Western General Hospital, Edinburgh EH4 2XU, UK. [50]Department of Engineering Science, University of Oxford,

Parks Road, Oxford OX1 3PJ, UK. [51]Wessex Clinical Genetics Service, University Hospital Southampton, Princess Anne Hospital, Coxford Road, Southampton SO16 5YA, UK. [52]Wessex Regional Genetics Laboratory, Salisbury NHS Foundation Trust, Salisbury District Hospital, Odstock Road, Salisbury, Wiltshire SP2 8BJ, UK. [53]Faculty of Medicine, University of Southampton, Building 85, Life Sciences Building, Highfield Campus, Southampton SO17 1BJ, UK. [54]South West Thames Regional Genetics Centre, St George's Healthcare NHS Trust, St George's, University of London, Cranmer Terrace, London SW17 0RE, UK. [55]Institute of Medical Genetics, University Hospital of Wales, Heath Park, Cardiff CF14 4XW, UK. [56]Department of Clinical Genetics, Block 12, Glan Clwyd Hospital, Rhyl, Denbighshire LL18 5UJ, UK. [57]East Anglian Medical Genetics Service, Box 134, Cambridge University Hospitals NHS Foundation Trust, Cambridge Biomedical Campus, Cambridge CB2 0QQ, UK. [58]Sheffield Regional Genetics Services, Sheffield Children's NHS Trust, Western Bank, Sheffield S10 2TH, UK. [59]Manchester Centre for Genomic Medicine, St Mary's Hospital, Central Manchester University Hospitals NHSFoundation Trust, Manchester Academic Health Science Centre, Manchester M13 9WL, UK. [60]North East Thames Regional Genetics Service, Great Ormond Street Hospital for Children NHS Foundation Trust, Great Ormond Street Hospital, Great Ormond Street, London WC1N3JH, UK. [61]North of Scotland Regional Genetics Service, NHS Grampian, Department of Medical Genetics Medical School, Foresterhill, Aberdeen AB25 2ZD, UK. [62]East of Scotland Regional Genetics Service, Human Genetics Unit, Pathology Department, NHS Tayside, Ninewells Hospital, Dundee DD1 9SY, UK. [63]Yorkshire Regional Genetics Service, Leeds Teaching Hospitals NHS Trust, Department of Clinical Genetics, Chapel Allerton Hospital, Chapeltown Road, Leeds LS7 4SA, UK. [64]North West Thames Regional Genetics Centre, North West London Hospitals NHS Trust, The Kennedy Galton Centre, Northwick Park and St Mark's NHS Trust Watford Road, Harrow HA1 3UJ, UK. [65]Oxford Regional Genetics Service, Oxford Radcliffe Hospitals NHS Trust, The Churchill Old Road, Oxford OX3 7LJ, UK. [66]West Midlands Regional Genetics Service, Birmingham Women's NHS Foundation Trust, Birmingham Women's Hospital, Edgbaston, Birmingham B15 2TG, UK. [67]Northern Genetics Service, Newcastle upon Tyne Hospitals NHS Foundation Trust, Institute of Human Genetics, International Centre for Life, Central Parkway, Newcastle upon Tyne NE1 3BZ, UK. [68]Northern Ireland Regional Genetics Centre, Belfast Health and Social Care Trust, Belfast City Hospital, Lisburn Road, Belfast BT9 7AB, UK. [69]Peninsula Clinical Genetics Service, Royal Devon and Exeter NHS Foundation Trust, Clinical Genetics Department, Royal Devon & Exeter Hospital (Heavitree), Gladstone Road, Exeter EX1 2ED, UK. [70]South East Thames Regional Genetics Centre, Guy's and St Thomas' NHS Foundation Trust, Guy's Hospital, Great Maze Pond, London SE1 9RT, UK. [71]Leicestershire Genetics Centre, University Hospitals of Leicester NHS Trust, Leicester Royal Infirmary (NHS Trust), Leicester LE1 5WW, UK. [72]Nottingham Regional Genetics Service, City Hospital Campus, Nottingham University Hospitals NHS Trust, The Gables, Hucknall Road, Nottingham NG5 1PB, UK. [73]West of Scotland Regional Genetics Service, NHS Greater Glasgow and Clyde, Institute of Medical Genetics, Yorkhill Hospital, Glasgow G3 8SJ, UK. [74]Bristol Genetics Service (Avon, Somerset, Gloucs and West Wilts), University Hospitals Bristol NHS Foundation Trust,  St Michael's Hospital, St Michael's Hill, Bristol BS2 8DT, UK. [75]Merseyside and Cheshire Genetics Service, Liverpool Women's NHS Foundation Trust, Department of Clinical Genetics, Royal Liverpool Children's Hospital Alder Hey, Eaton Road, Liverpool L12 2AP, UK. [76]National Centre for Medical Genetics, Our Lady's Children's Hospital, Crumlin, Dublin 12, Ireland. [77]Department of Clinical Genetics, Block 12, Glan Clwyd Hospital, Rhyl, Denbighshire LL18 5UJ Wales, UK. [78]Nuffield Department of Obstetrics & Gynaecology, University of Oxford, Level 3, Women's Centre, John Radcliffe Hospital, Oxford OX3 9DU, UK. [79]Institute of Biomedical Engineering, Department of Engineering Science, University of Oxford, Old Road Campus Research Building, Oxford OX3 7DQ, UK. [80]Big Data Institute, University of Oxford, Roosevelt drive, Oxford OX3 7LF, UK. [81]The Ethox Centre, Nuffield Department of Population Health, University of Oxford, Old Road Campus, Oxford OX3 7LF, UK

