## [Peer Review File · Nature Communications]

Reviewers' comments:

Reviewer #1 (Remarks to the Author):

The paper by Blok et al. describes the association of alterations in the chromatin remodelling gene CHD3 with intellectual disability, macrocephaly, and impaired speech and language. A de novo missense variant was initially found in a child with speech disorder with 35 additional cases found after surveying several cohorts that had undergone genomic sequencing. Most of the alterations impact the ATPase/helicase domain and functional analysis of several variants demonstrated an effect on ATPase activity. Overall this paper is well written and catalogues an impressive number of patients with CHD3 alterations making the association with a neurodevelopmental phenotype unequivocal. I only have a few points that should be addressed:

Major Points:

1. The biggest weakness of the paper is the equivocal results from the functional data and considering 3/5 of the constructs tested did not have significantly decreased ATP hydrolysis on nucleosome substrates, which is closer to what the native substrate would be in vivo (i.e. not naked DNA). In Supp. Fig 9 they mention that Trp1158 likely affects chromatin remodelling, so this is a good candidate to start with, and is right next to a residue they tested here and had little impact on nucleosome substrate. Because CHD3 is a histone deacetylase, why was a HDAC activity assay not performed? Published literature of other CHD genes has included such experiments including previous work but this group (HDAC1 interaction test) on CHD4(PMID: 27616479).
2. Is there a reason why the p.Trp1158Arg variant was not functionally tested given the potential its matching to a similar mutation in CHD4? It would seem to be one of the obvious variants to test.
3. There is at least one LOF variant identified that would be predicted to undergo NMD (p.Glu457*) and yet the authors also argue that it is unlikely that simple haploinsufficiency is the pathogenic mechanism for this gene (line 247). Can the authors expand on this, especially in light of the fact that although generally intolerant to LOF variants, there are several stop variants in EXAC.

Minor Points:

Line# 119-120- this statement is not exactly accurate, since in the following a few papers are cited that implicate autism and DD with variants in this gene. Moreover, Table 1 indicates that "autism" is not an uncommon feature of their patients (9/35). I would be better to soften this statement, perhaps it has not been associated in MIM or a comprehensive survey has not been completed.

In Figure 2B, only the splice site (green) and missense mutation (red) are shown in the legend. Please add the Blue LOF and organe (in frame).

Line #315 and #263 and various other locations- should the NM # have the version code associated with it? (i.e NM_001005273.2)

Figure 3 vs Figure S5- it is unclear if the % ATP hydrolyzed in Figure 3 is corrected for the baseline activity in figure S5.

Online methods line 266. Should be Illumina X Ten system. Please also add the version of BWA used.

Line #307- it would be helpful to note in the description of the mixing experiment in the main body of the submission that this was an equimolar mixing (I question whether there were serial dilution mixes, one vs many experiments, etc).

Figure S5 would benefit from having the scale on the Y axis reduced to 10% so it is easier to see differences in the activity.

In the supplementary notes, the section with the CHD3 model figures should be in the results, these do not seem to be methods.

Reviewer #2 (Remarks to the Author):

I have carefully reviewed the manuscript: "Missense mutations disrupting the helicase domain of chromatin remodeler CHD3 cause a novel neurodevelopmental syndrome with intellectual disability, macrocephaly and impaired speech and language."

The authors describe a relatively large cohort (35 patients) that all harbor de novo variants in the ATPase chromatin remodeler CHD3 and all have intellectual disability/developmental delay, speech delay/disorder, and some other overlapping phenotypes. In the cohort of 35 patients, the authors identify 23 unique mutations, all but four of which are missense. The authors perform a functional ATPase assays using recombinant CHD3 protein and discover disruption of ATPase activity in four of five tested patient mutations, however, one of these variants led to increased activity. Additionally, the authors performed computational structural analysis using two related proteins (structure for CHD3 is unavailable) to predict the location and consequence of patient mutations.

I think the manuscript is well written, timely and interesting to the community and I overall think that could fit for Nature Communications. Statistical analysis was appropriate and valid. In particular, this is a large set of patients, this is a brand new disorder and variants are de novo, and appear to change function and structure.

Major suggestions:

1) It is odd to have macrocephaly in the title since per table 1 it is only seen in 58% of individuals in the study. A more accurate title would be: "Missense mutations disrupting the helicase domain of chromatin remodeler CHD3 cause a novel neurodevelopmental syndrome with intellectual disability and impaired speech and language." This would help emphasize the major phenotypes that are seen in 100% of individuals.

2) The authors comment on the lack of significance of abnormalities in the functional assay for patient 29 (R1187P) and come up with a potential mechanism for why there was no effect in the assay for this particular variant. However, the variant from individual 3 (L915F) demonstrated higher activity in all the functional assays. The five variants tested appear to have been somewhat cherry picked by the authors (at least there is no rule given for how they were picked) and yet only 60% showed the proposed effect (loss of ATPase activity). The authors should comment on the results from this assay for patient 3. There are two possibilities for individual 3: 1) loss of function alleles and gain of function alleles lead to similar disease phenotypes. This is possible but hard to claim based on the data in this manuscript or 2) This particular allele is a false positive and this patient has another secondary cause for the disease phenotype. Given these data the authors should clearly state that their data suggests that this is not a purely ATPase dependent disorder and that some of the variants probably affect other aspects of CHD3 function.

3) The authors appear to have missed the fact that four of the six recently described patients with CHD1 variants also had speech apraxia, a relatively rare condition. I think this would be worth elaborating on in the manuscript since CHD1 and CHD3 share some domains (tandem chromodomains and SNF2-like ATPase domain) but are also discordant for others (paired PHD Zn-finger-like domains). Together, the data from these two papers make a compelling argument that dysfunctional chromatin remodeling leads to speech apraxia.

4)Page 12, line 247: The authors claim that the fact that there are no truncating variants argues for a dominant negative/gain of function mechanism. However, an alternative idea would be that patients with missense variants are hypomorphic but truncating variants are not compatible with survival. I feel the authors should be careful with any claims regarding specific mutational mechanism in this manuscript.

5) Page 12, line 254: Authors claim a "specific pattern of dysmorphic features". However on my examination of the individuals in the figures provided the dysmorphic features appear quite variable and the most common ones are relatively non-specific (widely spaced eyes and macrocephaly). I think it is unlikely that clinicians will use these features for triaging for CHD3 testing. Therefore, I think the authors should say in the manuscript that the facial phenotype is variable, but the speech apraxia is what should lead to interrogation of CHD3 as a possible cause of disease.

6) Authors claim that all the variants are de novo (page 8, line 165). This seems unlikely to me. Often it is very difficult to get parental samples and some parents refuse testing. Are there other patients that got left out of the manuscript in effort to ensure that only de novo variants were described? If so it would be nice to know the total number of variants found.

7) There are usually multiple variants discovered by exome sequencing. Authors claim that the CHD3 variant was always the most likely cause, however, I'd like to see the list of variants of unknown significance (additional supplementary table).

Minor suggestions:

1) In figure 2, it would be nice to label the functionally tested variants in some manner. In figure 2, there are only 34 variants shown, why not 35?

2) For figure 3 (figure S5&6), please plot individual points rather than plotting as bars. It could be nice to color the variants according to the colors used in Figure 2 to indicate the domains involved.

3)In supplemental S1, I'm unclear what the boxes denote. Please clarify.

4) The word mutation is used frequently (including the title), many entities are switching to variant (variant expected to change function, variant of unknown significance).

5) Table 1: has an inconsistent denominator, this is odd. I realize that some of the data may have been unavailable but there is no explanation given. Why not use a common denominator throughout the table or at least explain why not.

6)Page 1, line 2: Remodeller->remodeler. Similarly, remodelling->remodeling; I think this is a scientific word that should not vary by region of origin.

7) Page 6, line 97: "Unlike other members of CHD subfamily, pathogenic alterations": this sentence makes it seem that all other members of the CHD subfamily have associated with disease causing variant yet CHD5 has not been described at this time either, right?

8)Page 12, line 247: states 31 missense variants in the cohort but Page 8, line 165, states 23 different de novo variants. Also, we found 22 unique variants in figure 2. Please reconcile.

9) Page 15, line 327: aan->an

10) Page 17, line 359: organisation->organization

11)The authors should consider sharing the variants in a clinically relevant database (ClinVar or other similar).

12)It looks like the authors are not following all of the recommendations from the human genome variation society, however, journal should decide on how strict they want to be here since it can be a bit cumbersome to follow all the guidelines:

<http://varnomen.hgvs.org/recommendations/protein/variant/substitution/>

Overall, I think this is a nice manuscript that should be published. I hope my comments will help authors make the manuscript even better.

Kind regards,

Hans Tomas Bjornsson MD PhD
McKusick-Nathans Institute of Genetic Medicine
Johns Hopkins University, School of Medicine

Reviewer #3 (Remarks to the Author):

In this manuscript, the authors described the identification of a novel neurodevelopmental syndrome (Fig. 1) that is caused by mutations in the ATPase region of the CHD3 chromatin remodeling factor (Fig. 2). Recombinant proteins (Fig. S4) harboring some of these mutations exhibit variable levels of ATPase activity in the absence or presence of DNA or nucleosomes (Fig. 3; Fig. S5).

Although the general topic is interesting, the conclusions are not consistent with the data. These points need to be addressed.

Comments:

1. Some mutant proteins (e.g., R1121P and R1172Q) have lower ATPase activity than wild-type CHD3, and the L915F protein has higher ATPase activity than wild-type CHD3. In contrast, R1187P mutant CHD3 has essentially the same ATPase activity as wild-type CHD3. Therefore, the occurrence of the syndrome does not correlate with alterations in the ATPase activity.

2. The abstract states: "We analyzed the functional impact of several of the identified mutations and demonstrated that they affected ATPase activity". This statement is misleading and does not indicate that one mutant CHD3 protein does not have altered ATPase activity. Therefore, this sentence should be corrected.

3. It is possible that all of the mutant proteins are defective for an activity such as chromatin remodeling. At the least, the authors should test the chromatin remodeling activity of the wild-type and mutant proteins. Additional assays would further strengthen this work and possibly provide greater insight into the basis of the syndrome.

Response to reviewers' comments

We first would like to thank the editors and reviewers for their careful and fair review of our manuscript, and for the useful comments which have led to significant improvements of our manuscript. Below, please find a detailed reply to the specific comments of the reviewers.

Reviewer #1 (Remarks to the Author):

The paper by Blok et al. describes the association of alterations in the chromatin remodelling gene CHD3 with intellectual disability, macrocephaly, and impaired speech and language. A de novo missense variant was initially found in a child with speech disorder with 35 additional cases found after surveying several cohorts that had undergone genomic sequencing. Most of the alterations impact the ATPase/helicase domain and functional analysis of several variants demonstrated an effect on ATPase activity. Overall this paper is well written and catalogues an impressive number of patients with CHD3 alterations making the association with a neurodevelopmental phenotype unequivocal. I only have a few points that should be addressed:

Major Points:

1. The biggest weakness of the paper is the equivocal results from the functional data and considering 3/5 of the constructs tested did not have significantly decreased ATP hydrolysis on nucleosome substrates, which is closer to what the native substrate would be in vivo (i.e. not naked DNA). In Supp. Fig 9 they mention that Trp1158 likely affects chromatin remodelling, so this is a good candidate to start with, and is right next to a residue they tested here and had little impact on nucleosome substrate. Because CHD3 is a histone deacetylase, why was a HDAC activity assay not performed? Published literature of other CHD genes has included such experiments including previous work but this group (HDAC1 interaction test) on CHD4(PMID: 27616479)).

We agree with the reviewer that the functional investigations of the original submission had some limitations. To address this point and improve our understanding of the pathogenic mechanisms underlying this disorder, we have carried out substantive additional testing with complementary methods. In particular, we performed new assays directly targeting chromatin remodeling, which is the main molecular function of the protein. Moreover, following the recommendation of the reviewer we not only investigated the original set of mutations, but also added p.Trp1158Arg to our assays, to specifically target that critical residue.

For this additional functional work, we focused specifically on remodeling rather than HDAC activity, because the locations of the mutations in our cohort suggest effects on the core function of the CHD3 protein: chromatin remodeling. CHD3 is not a histone deacetylase itself, but can exert its chromatin remodeling effects within the NuRD complex, in which HDAC1 and HDAC2 function as histone deacetylases. The interaction of CHD3 with HDACs is thought to occur via a conserved motif at the C-terminal end of the CHD3 protein. It is therefore unlikely that missense mutations in the helicase domain affect the interaction with HDACs. In addition, in the previous CHD4 work that the reviewer cites, there was no disturbance of CHD4-HDAC1 interaction for the tested mutants. This could be an interesting topic for future investigation, especially with regard to the few C-terminal mutations outside the helicase domain, but it is beyond the scope of the current study and is not an issue affecting the primary conclusions of our paper.

2. Is there a reason why the p.Trp1158Arg variant was not functionally tested given the potential its matching to a similar mutation in CHD4? It would seem to be one of the obvious variants to test.

This variant was not originally tested with functional assays because the patient was added to our cohort after this part of the study was first established. We agree that the p.Trp1158Arg is an important variant to test, and therefore we have now included it in the functional experiments, both for the original ATP-ase assays and the newer chromatin remodeling tests.

3. There is at least one LOF variant identified that would be predicted to undergo NMD (p.Glu457*) and yet the authors also argue that it is unlikely that simple haploinsufficiency is the pathogenic mechanism for this gene (line 247). Can the authors expand on this, especially in light of the fact that although generally intolerant to LOF variants, there are several stop variants in EXAC.

This is indeed an intriguing point. If we look at the data in the ExAC database in more detail (data from 5-2-2018), there are 16 different LoF variants. Each of these variants is only found once, with the exception of one variant (g.7814942C/T) that is found three times:

All Missense + LoF LoF Include filtered (non-PASS) variants

Export table to CSV

† denotes a consequence that is for a non-canonical transcript

Variant	Chrom	Position	Consequence	Filter	Annotation	Flags	Allele Count	Allele Number
17:7792319 A / AT	17	7792319	p.Met17†	PASS	frameshift		1	121384
17:7792373 AG / A	17	7792373	p.Ile20PhefsTer28†	PASS	frameshift		1	121408
17:7793893 G / A	17	7793893	n.60-1G>A†	PASS	splice acceptor	LC LoF	1	68422
17:7794256 A / T	17	7794256	c.562-2A>T	PASS	splice acceptor		1	121326
17:7797122 G / T	17	7797122	c.971-1G>T	PASS	splice acceptor		1	121174
17:7797731 A / C	17	7797731	c.1253-2A>C	PASS	splice acceptor		1	118190
17:7798239 A / AG	17	7798239	p.Glu485GlyfsTer29	PASS	frameshift		1	118950
17:7798282 C / G	17	7798282	p.Tyr498Ter	PASS	stop gained		1	121148
17:7798795 C / T	17	7798795	p.Arg607Ter	PASS	stop gained		1	121410
17:7806671 C / T	17	7806671	p.Arg1252Ter	PASS	stop gained		1	121334
17:7807449 G / GA	17	7807449	p.Arg50LysfsTer9†	PASS	frameshift	LC LoF	1	14430
17:7811208 G / C	17	7811208	c.8-1G>C†	PASS	splice acceptor		1	117640
17:7811209 CAGA / C	17	7811209	c.5205_5207delAGA	PASS	splice acceptor		1	117804
17:7811274 C / T	17	7811274	p.Arg1756Ter	PASS	stop gained		1	120950
17:7814897 C / T	17	7814897	p.Arg343Ter†	PASS	stop gained		1	120918
17:7814942 C / T	17	7814942	p.Arg358Ter†	PASS	stop gained		3	112780

Some variants are annotated as possible splice variants, and others are only expected to yield LoF consequence in a non-canonical transcript. Therefore, we checked the effects of all variants on two different CHD3 transcripts manually using our variant analysis software (Alamut). The table below gives the results of this analysis.

hg19	Effect (NM_001005271)	Effect (NM_001005273.2)	Allele count in ExAC
17:7792319 A/AT	Intron	Start loss	1
17:7792373 AG/A	Intron	Frameshift (120Ffs)	1
17:1193893 G/A	Missense	Missense + possible splice (weak)	1
17:7794256 A/T	Possible splice	Possible splice	1
17:7797122 G/T	Possible splice (in frame deletion of 2 a.a.)	Possible splice (in frame deletion of 2 a.a.)	1
17:7797731 A/C	Possible splice (very weak)	Possible splice (very weak)	1
17:7798239 A/AG	Frameshift	Frameshift	1

17:7798282 C/G	Nonsense (Y498*)	Nonsense (Y439*)	1
17:7798795 C/T	Nonsense (R607*)	Nonsense (R548*)	1
17:7806671 C/T	Nonsense (R1252*)	Nonsense (R1193*)	1
17:7807449 G/GA	Intron	Intron	1
17:7811208 G/C	Intron (possible splice)	Intron (possible splice)	1
17:7811209 CAGA/C	Deletion of 1 residue (Q1735del)	Deletion of 1 residue (Q1676del)	1
17:7811274 C/T	Nonsense (R1756*)	Nonsense (R1697*)	1
17:7814897 C/T	Synonymous	Synonymous	1
17:7814942 C/T	3'UTR substitution	3'UTR substitution	1

Table: Interpretation of possible LoF variants in ExAC.

There are two main transcripts described for CHD3. NM_001005271 is the longest transcript, encoding a protein of 2059 amino acids. This is the canonical transcript according to ExAC, and is most highly expressed according to GTEX data. The other transcript, NM_001005273.2, encodes a protein of 2000 amino acids and is the canonical transcript according to Uniprot. This is the transcript we have used to annotate our functional data and manuscript. The difference between these two transcripts is the presence/absence of one exon at the beginning of the gene.

As the table above shows, ExAC actually contains only **five clear LoF mutations** (nonsense/frameshift effects for both the alternative transcripts of CHD3), and each has an allele count of 1. This is an extremely low number, considering the large size of the gene, and the total number of alleles present in ExAC (~120,000 alleles). Thus, consistent with the text of our manuscript, this gene is extremely intolerant for LoF mutations, which is also seen in the ExAC LoF-intolerance score (pLI) of 1.0.

We hypothesize that the presence of a few (as opposed to no) LoF mutations in ExAC can be explained by:

1. Potential presence of mild neurodevelopmental disorders in the ExAC cohort (e.g. mild/borderline ID), given that it includes over 50,000 individuals.
2. Mosaicism for some of the mutations (i.e. present in only certain tissues such as blood)
3. Other molecular 'escape mechanisms', e.g. reduced penetrance or variable expression

All in all, we do not doubt the pathogenicity of the CHD3 nonsense mutation that is expected to undergo NMD in our cohort. This hypothesis is strengthened by the overlapping phenotype (including macrocephaly) and the relative absence of these mutations in ExAC (pLI of 1.0). CHD3 also has a Haploinsufficiency Score (HI Index) of 16.92%, suggesting haploinsufficiency is deleterious¹. Unfortunately, it is not possible to obtain phenotypic information on specific ExAC individuals to confirm our hypothesis.

Minor Points:

Line# 119-120- this statement is not exactly accurate, since in the following a few papers are cited that implicate autism and DD with variants in this gene. Moreover, Table 1 indicates that "autism" is not an uncommon feature of their patients (9/35). I would be better to soften this statement, perhaps it has not been associated in MIM or a comprehensive survey has not been completed.

The statement from the original manuscript that this comment refers to is: *"In contrast to most other members of the CHD protein family, mutations in CHD3 have not yet been implicated in a human disorder"*. In large trio exome studies, with now more than 50,000 trios published, it is possible to find papers showing de novo mutations in almost any of the 22,000 genes of our genome. However, no prior study has focused on CHD3, performed functional characterization of the variants, or to provide a phenotypic profile of the affected patients. Nonetheless, we agree that this statement

should be softened, to avoid misunderstanding. Therefore, in the revised manuscript we changed it into:

“A specific syndrome associated with CHD3 mutations has not yet been characterized.”

In Figure 2B, only the splice site (green) and missense mutation (red) are shown in the legend. Please add the Blue LOF and organe (in frame).

We have adjusted this in the revised manuscript.

Line #315 and #263 and various other locations- should the NM # have the version code associated with it? (i.e NM_001005273.2)

We have adjusted this in the revised manuscript, so that the complete NM number including the version code is now listed throughout.

Figure 3 vs Figure S5- it is unclear if the % ATP hydrolyzed in Figure 3 is corrected for the baseline activity in figure S5.

The % ATP hydrolyzed is in the revised manuscript corrected for the wild-type amount of % ATP hydrolyzed. It is not corrected for the baseline activity in figure S5. The uncorrected (raw) values of the ATPase assays are now added as a Supplemental Table 2. We have also added a sentence to the legends of Figure 3 and Figure S5 to explain the normalization of the ATP hydrolysis data.

Online methods line 266. Should be Illumina X Ten system. Please also add the version of BWA used.

We have adjusted this and added the version number of the BWA.

Line #307- it would be helpful to note in the description of the mixing experiment in the main body of the submission that this was an equimolar mixing (I question whether there were serial dilution mixes, one vs many experiments, etc).

We have adjusted this in the revised manuscript.:

*“To assess the possibility of a dominant-negative effect of the mutations, we mixed wild-type protein with **equimolar amounts of** each of the five mutant proteins, but found only additive effects in these experiments (Figure S6).”*

Figure S5 would benefit from having the scale on the Y axis reduced to 10% so it is easier to see differences in the activity.

We have adjusted this.

In the supplementary notes, the section with the CHD3 model figures should be in the results, these do not seem to be methods.

We agree and have moved this section to the Results section of the revised manuscript.

Reviewer #2 (Remarks to the Author):

I have carefully reviewed the manuscript: "Missense mutations disrupting the helicase domain of chromatin remodeler CHD3 cause a novel neurodevelopmental syndrome with intellectual disability, macrocephaly and impaired speech and language."

The authors describe a relatively large cohort (35 patients) that all harbor de novo variants in the ATPase chromatin remodeler CHD3 and all have intellectual disability/developmental delay, speech delay/disorder, and some other overlapping phenotypes. In the cohort of 35 patients, the authors identify 23 unique mutations, all but four of which are missense. The authors perform a functional ATPase assays using recombinant CHD3 protein and discover disruption of ATPase activity in four of five tested patient mutations, however, one of these variants led to increased activity. Additionally, the authors performed computational structural analysis using two related proteins (structure for CHD3 is unavailable) to predict the location and consequence of patient mutations.

I think the manuscript is well written, timely and interesting to the community and I overall think that could fit for Nature Communications. Statistical analysis was appropriate and valid. In particular, this is a large set of patients, this is a brand new disorder and variants are de novo, and appear to change function and structure.

Major suggestions:

1) It is odd to have macrocephaly in the title since per table 1 it is only seen in 58% of individuals in the study. A more accurate title would be: "Missense mutations disrupting the helicase domain of chromatin remodeler CHD3 cause a novel neurodevelopmental syndrome with intellectual disability and impaired speech and language." This would help emphasize the major phenotypes that are seen in 100% of individuals.

It is true that macrocephaly is only seen in 58% of the individuals. As macrocephaly (a head circumference that is at least 2.5 SD above the mean value for a given age and gender) is a very rare phenomenon, only present in 0.6% of the population, we view it as a key feature of the phenotypic spectrum of this CHD3-related disorder, and think it is important to mention this in the article and in the title.

2) The authors comment on the lack of significance of abnormalities in the functional assay for patient 29 (R1187P) and come up with a potential mechanism for why there was no effect in the assay for this particular variant. However, the variant from individual 3 (L915F) demonstrated higher activity in all the functional assays. The five variants tested appear to have been somewhat cherry picked by the authors (at least there is no rule given for how they were picked) and yet only 60% showed the proposed effect (loss of ATPase activity). The authors should comment on the results from this assay for patient 3. There are two possibilities for individual 3: 1) loss of function alleles and gain of function alleles lead to similar disease phenotypes. This is possible but hard to claim based on the data in this manuscript or 2) This particular allele is a false positive and this patient has another secondary cause for the disease phenotype. Given these data the authors should clearly state that their data suggests that this is not a purely ATPase dependent disorder and that some of the variants probably affect other aspects of CHD3 function.

We agree with the reviewer that the ATPase activity assays did not show a similar result for all mutations. As addressed in the reply to comment 1 of Reviewer 1 as well, we added a chromatin remodeling assay to our study, as a disturbance of ATPase activity is one but definitely not the only possible pathogenic mechanism for the mutations. Moreover, CHD3 makes use of the ATPase activity to induce a conformational change in order to perform chromatin remodeling. Using this remodeling assay, we show that some mutations that preserve ATPase activity show very obvious disruptions in

chromatin remodeling assays. This is perfectly in line with evidence from a recent study on a similar ATP-dependent chromatin remodeler (SNF2)², in which a tryptophan residue at a position analogous to CHD3 residue Trp1158 is shown to be critical for chromatin remodeling, but not for ATP hydrolysis.

The p.Leu925Phe mutation that showed an increased ATPase activity has a similar overactive effect in the chromatin remodeling assay. We propose in the revised manuscript that a perturbation of chromatin remodeling activity of CHD3, whether this is a gain or loss of activity compared to wild-type, might both lead to disturbance of the chromatin landscape during neurodevelopment and subsequently to a neurodevelopmental phenotype. A similar mechanism has recently been shown for cancer-specific mutations in SMARCA4³, we refer to this study in our revised manuscript as well.

The p.Arg1187Pro mutation shows no difference in ATPase and chromatin remodeling activity compared to wild-type CHD3 in our assays. We do not have a clear explanation for this. It could be that the pathogenic mechanism is qualitatively different for this mutant compared to the other mutants. Alternatively, the effect size of this mutation might be beyond the range of our assays. In theory, the p.Arg1187Pro mutation could also be a normal (non-pathogenic) variant. Considering the fact that we have two unrelated individuals in our cohort with this mutation and a very similar phenotype compared to the rest of the cohort (including macrocephaly), we do not think this is very likely. We consider it is more likely that the mutation affects CHD3 or its complexes in a manner we did not yet assess, for example by affecting its affinity to other proteins, or the function of its interactors.

3) The authors appear to have missed the fact that four of the six recently described patients with CHD1 variants also had speech apraxia, a relatively rare condition. I think this would be worth elaborating on in the manuscript since CHD1 and CHD3 share some domains (tandem chromodomains and SNF2-like ATPase domain) but are also discordant for others (paired PHD Zn-finger-like domains). Together, the data from these two papers make a compelling argument that dysfunctional chromatin remodeling leads to speech apraxia.

We thank the reviewer for this excellent suggestion. We have added explicit discussion of these important issues in the revised manuscript (page 10, lines 13-17).

4)Page 12, line 247: The authors claim that the fact that there are no truncating variants argues for a dominant negative/gain of function mechanism. However, an alternative idea would be that patients with missense variants are hypomorphic but truncating variants are not compatible with survival. I feel the authors should be careful with any claims regarding specific mutational mechanism in this manuscript.

We think this is a very important issue, and we still do not know the exact details on a mechanistic level for the missense and truncating variants. See also our answer to comment 3 of reviewer 1. We did indeed consider the hypothesis suggested by reviewer 2: that missense mutations are hypomorphic while truncating mutations are lethal. But the following findings do not support that view:

- Truncating alleles are present (albeit at very low frequency) in the ExAC database.
- There is a patient in our cohort with a nonsense mutation that is predicted to undergo NMD.
- Deletions spanning CHD3 are present in the Decipher database.

We agree that we should be careful with claims regarding specific mutational mechanisms. However, we feel that the part of the discussion that the reviewer refers to does not include any strong claim on this topic. Although we present some data that fits best in a non-haploinsufficiency mechanism, we do not have a good handle on the precise pathogenic mechanism behind it.

5) Page 12, line 254: Authors claim a “specific pattern of dysmorphic features”. However on my examination of the individuals in the figures provided the dysmorphic features appear quite variable and the most common ones are relatively non-specific (widely spaced eyes and macrocephaly). I think it is unlikely that clinicians will use these features for triaging for CHD3 testing. Therefore, I think the authors should say in the manuscript that the facial phenotype is variable, but the speech apraxia is what should lead to interrogation of CHD3 as a possible cause of disease.

It is indeed unlikely for clinicians to use the features to specifically test for CHD3-associated disorder, and we agree on the variability of the facial phenotype. Nowadays, with so many new conditions identified, and milder and more severe ends of the spectrum of diseases being defined, it is hard to pick a candidate gene from examining a child with ID and minor dysmorphisms, unless it is a classic recognizable syndrome. Still, it is important to keep striving to establish recognizable features, and we consider that there is a clear pattern of malformations and dysmorphisms here. As the facial phenotype is not completely similar for all patients and not always recognizable, we have removed the word ‘specific’ from the claim above.

In addition to that, the speech apraxia is also not present in all patients, so we think it should be the combination of different phenotypic features that might lead to recognition of this disorder, and if not recognized, it might help later in the interpretation of unclear WES/WGS variants as well.

6) Authors claim that all the variants are *de novo* (page 8, line 165). This seems unlikely to me. Often it is very difficult to get parental samples and some parents refuse testing. Are there other patients that got left out of the manuscript in effort to ensure that only *de novo* variants were described? If so it would be nice to know the total number of variants found.

We specifically searched in our databases for *de novo* variants. All variants in our manuscript are confirmed to be *de novo*. We did not exclude any patients from our study, but in general, geneticists connect with each other for candidate genes once they know a variant is *de novo*.

7) There are usually multiple variants discovered by exome sequencing. Authors claim that the CHD3 variant was always the most likely cause, however, I’d like to see the list of variants of unknown significance (additional supplementary table).

We asked all co-authors to provide details on additional variants of unknown significance that were considered when analyzing the whole exome or genome sequencing data. Although many co-authors could provide us with these data for the patients that they included, for some centers it was not possible to share information on additional variants because of institutional regulations. Finally, some co-authors only discussed the best candidate gene (CHD3) with the families, and do not wish for all VUSs to be published for their patient.

Nevertheless, for the reviewers to have access to the same data as we did, we added a table with details on additional single nucleotide variants and copy number variation found in the individuals in our cohort. This table is for review only and not for publication (Table A: Additional Variants).

The table shows the single nucleotide variants and copy number variations reported in our cohort. With the exception of a *de novo* truncating variant in the CIC-gene, there are no additional variants reported that are likely to contribute to the neurodevelopmental phenotype of the individuals in our cohort. The respective CIC-variant was already mentioned in our supplementary table with phenotypic and mutation details on all individuals (Supplementary Table 1). We have now added a mention of this *de novo* variant in the main text: “*In proband 15 who has a de novo CHD3 p.Asp1120His mutation, a de novo truncating mutation in CIC was also identified*”

(NM_015125.3:c.1444G>T; p.Glu482). Since truncating mutations in CIC were recently suggested as a potential cause of intellectual disability (ID), both mutations might be involved in the phenotype of this proband.”*

Minor suggestions:

1) In figure 2, it would be nice to label the functionally tested variants in some manner. In figure 2, there are only 34 variants shown, why not 35?

We think this is a great idea and have labelled the functionally tested variants in the revised figure. It is true that one variant was missing from the figure (p.Thr1136Ile), and this is now corrected in the revised manuscript.

2) For figure 3 (figure S5&6), please plot individual points rather than plotting as bars. It could be nice to color the variants according to the colors used in Figure 2 to indicate the domains involved.

We have adjusted the graphs, and plotted individual data points as well as bars indicating the average values. We also liked the idea of colouring the variants according to the domains involved, but as we have changed the layout of our graph in Figure 3 (and as all variants except one are located in the Helicase C-terminal domain) we think it gets too confusing to present the data this way.

3) In supplemental S1, I'm unclear what the boxes denote. Please clarify.

The boxes show the different conserved SNF2 motifs (motif I, Ia, II, III, IV, V and VI). In the revised manuscript we have added a clarification to the legend of Figure S1.

4) The word mutation is used frequently (including the title), many entities are switching to variant (variant expected to change function, variant of unknown significance).

The reviewer is right about the use of 'mutation' and 'variant' throughout our manuscript. As the main conclusion of our paper is the fact that the CHD3 variants are pathogenic, we have chosen in the revised manuscript to use 'mutation' instead of 'variant'. In order to achieve consistency, we changed the word 'variant' in the manuscript into 'mutation' for all the mutations found in our cohort.

5) Table 1: has an inconsistent denominator, this is odd. I realize that some of the data may have been unavailable but there is no explanation given. Why not use a common denominator throughout the table or at least explain why not.

We have added an explanation for this underneath the table:

As information on the different features was not always applicable or known for each patient, the denominator in the 'Amount' column is different for different clinical characteristics.

6) Page 1, line 2: Remodeller->remodeler. Similarly, remodelling->remodeling; I think this is a scientific word that should not vary by region of origin.

We have adjusted this throughout the revised manuscript.

7) Page 6, line 97: "Unlike other members of CHD subfamily, pathogenic alterations": this sentence makes it seem that all other members of the CHD subfamily have associated with disease causing variant yet CHD5 has not been described at this time either, right?

This is true and it might be misleading. We have adjusted it in the revised manuscript into “Unlike most other members of the CHD subfamily, ...”

8)Page 12, line 247: states 31 missense variants in the cohort but Page 8, line 165, states 23 different de novo variants. Also, we found 22 unique variants in figure 2. Please reconcile.

- There are 31 patients with missense mutations in our cohort. We have adjusted the sentence in page 12, line 247 to:
“The paucity of **patients with** truncating mutations compared to the 31 **patients with** missense mutations in our cohort also supports this view, ...”
- There are 23 different de novo variants. Accidentally, figure 2 missed 1 missense mutation. We have now adjusted this.

9) Page 15, line 327: aan->an

This is adjusted.

10) Page 17, line 359: organisation->organization

This is adjusted.

11)The authors should consider sharing the variants in a clinically relevant database (ClinVar or other similar).

We completely agree that we should share the data in Clinvar. We have submitted the mutations of our study to ClinVar (submission ID SUB3976460), and the data will be released once the manuscript is published.

12)It looks like the authors are not following all of the recommendations from the human genome variation society, however, journal should decide on how strict they want to be here since it can be a bit cumbersome to follow all the guidelines:

<http://varnomen.hgvs.org/recommendations/protein/variant/substitution/>

We have tried to use the HGVS nomenclature consistently throughout the revised manuscript. However, if the journal wants us to change nomenclature, we are happy to adjust to the journal guidelines.

Overall, I think this is a nice manuscript that should be published. I hope my comments will help authors make the manuscript even better.

Kind regards,

Hans Tomas Bjornsson MD PhD
McKusick-Nathans Institute of Genetic Medicine
Johns Hopkins University, School of Medicine

Reviewer #3 (Remarks to the Author):

In this manuscript, the authors described the identification of a novel neurodevelopmental syndrome (Fig. 1) that is caused by mutations in the ATPase region of the CHD3 chromatin remodeling factor (Fig. 2). Recombinant proteins (Fig. S4) harboring some of these mutations exhibit variable levels of ATPase activity in the absence or presence of DNA or nucleosomes (Fig. 3; Fig. S5).

Although the general topic is interesting, the conclusions are not consistent with the data. These points need to be addressed.

Comments:

1. Some mutant proteins (e.g., R1121P and R1172Q) have lower ATPase activity than wild-type CHD3, and the L915F protein has higher ATPase activity than wild-type CHD3. In contrast, R1187P mutant CHD3 has essentially the same ATPase activity as wild-type CHD3. Therefore, the occurrence of the syndrome does not correlate with alterations in the ATPase activity.

We agree that the ATPase data did not show an identical pattern for all mutations. CHD3 hydrolyzes ATP and couples this to chromatin remodeling, and some mutations might disturb the protein functions in a different way. As also noted in our response to reviewer 1, we performed substantive new assays (restriction enzyme accessibility assay) to test the remodeling capacity of the different mutant proteins. For the mutations that had disturbed ATPase activity, the remodeling assay showed a very similar effect, supporting the findings of the ATPase assays. This is also what would be expected, as the remodeling function of the CHD3 protein is dependent of the ATPase activity. Some mutations that did not show a clear decrease in ATPase activity (p.Asn1159Lys and the newly added p.Trp1158Arg), have dramatically reduced remodeling capacities. This is also in line with the expectations, considering the location of these mutations in the three-dimensional protein structure and the information about this specific position in context to chromatin remodeling from a study on SNF2², a similar ATP-dependent chromatin remodeler. The results of the added chromatin remodeling assay clearly show how chromatin remodeling can be influenced by a disturbed ATPase activity, but can also be affected independently of the ATPase function. As also written in our reply on comment 2 of Reviewer 2, we do not know why the p.Arg1187Pro mutation does not show any effect in both our functional assays, but hypothesize it may affect CHD3 interactions, as we now mention in the text.

All in all, we think that significant alterations in chromatin remodeling in either direction (whether overactive or underactive) can cause the phenotypes in this neurodevelopmental disorder, and discuss this in more detail in the revised manuscript.

2. The abstract states: "We analyzed the functional impact of several of the identified mutations and demonstrated that they affected ATPase activity". This statement is misleading and does not indicate that one mutant CHD3 protein does not have altered ATPase activity. Therefore, this sentence should be corrected.

We agree with this, and have corrected it in the revised manuscript, to: *"Functional analysis of a subset of the identified mutations revealed alterations in chromatin remodeling properties"*

3. It is possible that all of the mutant proteins are defective for an activity such as chromatin remodeling. At the least, the authors should test the chromatin remodeling activity of the wild-type and mutant proteins. Additional assays would further strengthen this work and possibly provide greater insight into the basis of the syndrome.

We thank the reviewer for this insightful suggestion. As recommended, we have now tested nucleosome remodeling function of the original panel of mutant proteins (and included one additional mutant – Trp1158Arg). As noted by the reviewer, the additional assays have strengthened the study and provided greater insights. We now provide details in the text and discussion consistent with the notion that alterations in chromatin remodeling (both gain and loss of function) are likely molecular consequences leading to pathology.

References

1. Huang, N., Lee, I., Marcotte, E.M. & Hurles, M.E. Characterising and predicting haploinsufficiency in the human genome. *PLoS Genet* **6**, e1001154 (2010).
2. Liu, X., Li, M., Xia, X., Li, X. & Chen, Z. Mechanism of chromatin remodelling revealed by the Snf2-nucleosome structure. *Nature* **544**, 440-445 (2017).
3. Hodges, H.C. *et al.* Dominant-negative SMARCA4 mutants alter the accessibility landscape of tissue-unrestricted enhancers. *Nat Struct Mol Biol* **25**, 61-72 (2018).

REVIEWERS' COMMENTS:

Reviewer #1 (Remarks to the Author):

The authors have addressed my comments and the manuscript has been strengthened. The addition of complementary functional methods/data, despite the limitations, is appreciated.

Reviewer #2 (Remarks to the Author):

I feel that the authors have responded appropriately to my comments. I suggest that manuscript be accepted for publication.

Reviewer #3 (Remarks to the Author):

In the revised manuscript, the authors have performed the requested chromatin remodeling experiments and revised the text accordingly. Publication is recommended.

Minor point:

In Fig. 4B, the Y axis is "% cut", but the numbers are given as fractions instead of percentages. For instance, "0.2" should be "20".